# An artificial metalloenzyme biosensor can detect ethylene gas in fruits and Arabidopsis leaves

Kenward Vong [1,7], Shohei Eda[1,2,7], Yasuhiro Kadota [3], Igor Nasibullin [1], Takanori Wakatake[3], Satoshi Yokoshima[4], Ken Shirasu [3] & Katsunori Tanaka [1,2,5,6]*

Enzyme biosensors are useful tools that can monitor rapid changes in metabolite levels in real-time. However, current approaches are largely constrained to metabolites within a limited chemical space. With the rising development of artificial metalloenzymes (ArM), a unique opportunity exists to design biosensors from the ground-up for metabolites that are difficult to detect using current technologies. Here we present the design and development of the ArM ethylene probe (**AEP**), where an albumin scaffold is used to solubilize and protect a quenched ruthenium catalyst. In the presence of the phytohormone ethylene, cross metathesis can occur to produce fluorescence. The probe can be used to detect both exogenous- and endogenous-induced changes to ethylene biosynthesis in fruits and leaves. Overall, this work represents an example of an ArM biosensor, designed specifically for the spatial and temporal detection of a biological metabolite previously not accessible using enzyme biosensors.

[1] Biofunctional Synthetic Chemistry Laboratory, RIKEN Cluster for Pioneering Research, 2-1 Hirosawa, Wako-shi, Saitama 351-0198, Japan. [2] GlycoTargeting Research Laboratory, RIKEN Baton Zone Program, 2-1 Hirosawa, Wako-shi, Saitama 351-0198, Japan. [3] Plant Immunity Research Group, RIKEN Center for Sustainable Resource Science, 1-7-22 Suehiro-cho, Tsurumi, Yokohama, Kanagawa 230-0045, Japan. [4] Graduate School of Pharmaceutical Sciences, Nagoya University, Furo-cho, Chikusa-ku, Nagoya 464-8601, Japan. [5] Biofunctional Chemistry Laboratory, A. Butlerov Institute of Chemistry, Kazan Federal University, 18 Kremlyovskaya street, Kazan 420008, Russia. [6] Department of Chemical Science and Engineering, School of Materials and Chemical Technology, Tokyo Institute of Technology, 2-12-1 O-okayama, Meguro-ku, Tokyo 152-8552, Japan. [7] These authors contributed equally: Kenward Vong, Shohei Eda. *email: kotzenori@riken.jp

Enzyme-based chemical biosensors for diagnostic purposes still remains an underdeveloped research area. One of the major issues impeding progress is that most, if not all, enzyme biosensors are derived from existing natural enzymes, rather than methodically designed. As such, there is still a large chemical space that remains inaccessible based on current developmental approaches to enzyme biosensors.

One challenging metabolite to detect is ethylene (IUPAC name: ethene), a well-characterized phytohormone that plays a major role in regulating many aspects of plant growth, immunity, development, and senescence[1,2]. In literature, two systems are known to control ethylene biosynthesis in plants[3]. System 1 production, which is ethylene autoinhibitory[4], is applicable to all plants and is mainly responsible for controlling basal ethylene levels for normal vegetative growth. Under environmental or developmental stimuli, ethylene levels in climacteric fruit can dramatically increase via system 2 production, which is an ethylene autocatalytic process[5–7]. In literature, exogenous sources of ethylene have been shown to greatly accelerate the processes of abscission and ripening[8–11].

Given its biological importance, improved diagnostic tools for ethylene gas would be invaluable to agricultural research to help better study ethylene-dependent pathways during stress response, hormone treatment, and different developmental stages in plants. However, since ethylene is a small gaseous olefin that freely diffuses between membranes under physiological conditions, there are many obstacles to overcome in order to develop working systems. Current methods to assay ethylene in plants have largely employed analytical techniques (such as gas chromatography, electrochemical sensors, and laser-based techniques[12]), or have utilized plants genetically encoded with ethylene-inducible reporters (such as EBS:GUS[13,14]). As an alternative method for ethylene detection, interest has recently begun to shift towards developing chemical-based probes. Through this approach, topical application onto plant tissues should in theory allow ethylene levels to be monitored in a spatiotemporal manner.

In one of the earliest examples of a chemical-based ethylene probe, the Burstyn group developed poly(vinyl phenyl ketone) (PVPK) polymer films. In the presence of ethylene, coordination with impregnated silver ions could lead to the proportional decrease of photoluminescence[15]. Later, the Swager group developed an alternative system where the quenched fluorescence of conjugated polymers by copper(I) scorpionate complexes could be revived by displacement with ethylene[16]. Another notable example is from the Kodera group, where a silver(I)–anthracene complex was constructed around a crucial metal–arene interaction. Upon exposure to ethylene, interconversion from a metal–arene to metal–ethylene interaction led to a change in the anthracene absorption and fluorescence spectra[17]. Finally, during the preparation of this manuscript, two reports were published highlighting an elegant strategy using profluorescent ruthenium metal complexes[18,19]. In the presence of ethylene, turn-on fluorescence can be triggered via cross metathesis to release a BODIPY-based dye. In particular, the Michel group showed the detection of diffused ethylene from live cell cultures incubated with ripening fruit[18]. Although a clear upgrade in terms of biocompatibility compared to previous strategies, the direct imaging of fruits or plants was not shown in this report.

Among all current examples of ethylene-sensing probes, one common theme is the use of water-insoluble metal catalysts. For instance, besides the Michel report, all studies have heavily relied on the use of organic solvent. Furthermore, none of these methods have been able to test ethylene directly from plant tissue; instead relying on exogenous ethylene sources. It can be surmised that in the presence of water-based plant samples, metal complexes may aggregate, potentially lowering their reactivity. In addition, metal quenching in complex biological environments is also an issue. Numerous reports in literature have identified high concentrations of glutathione in plants[20]. As a result, for metals like ruthenium, inactivation by low molecular weight thiols becomes an added challenge. In order to overcome these practical limitations, what is currently lacking is a means to confer biocompatibility to metal complex catalysts.

Artificial metalloenzymes (ArM) are the result of incorporating abiotic metals complexes into a protein scaffold, which can often impart enhanced biofunctionality to the anchored metal catalysts. In current literature, a host of different protein scaffolds have been adapted as ArMs, some of which include streptavidin[21–26], prolyl oligopeptidase[27,28], myoglobin[29,30], CeuE[31], ADORA2A[32], nitrobindin[33], among many others. Recently, our group explored the use of human serum albumin-based ArMs, which was found to impart two important biological effects for the anchored metal catalyst[34]. First, the typically insoluble ruthenium complex catalyst (linked to a coumarin-ligand anchor) can become soluble in aqueous conditions with the addition of albumin. This is likely the effect of the hydrophobic-binding pocket on albumin, which is well known to bind to many hydrophobic drug molecules[35]. And second, due to the negatively charged surface of albumin at physiological pH[36], charged metabolites were found to be prevented from entering the hydrophobic-binding pocket. As such, metal catalysts (e.g. ruthenium) bound inside the hydrophobic-binding pocket could be resistant to inactivation from biological thiols (i.e. glutathione). To highlight the biocompatibility of albumin-based ArMs bound with ruthenium (alb–Ru), investigations were then further done to develop an anticancer prodrug therapy based on activation via ring-closing metastasis (RCM)[34]. In this case, alb–Ru was furnished with cancer cell-targeting complex N-glycans[37–41], which was then used to successfully activate a RCM-drug precursor against various cancer lines (e.g. HeLa, A549, SW620).

Not only restricted to drug therapy, the chemoselective nature of abiotic metal-containing ArMs can also open the door for enzyme biosensor applications. Although metal-based diagnostic probes have been well studied in literature, many of them do not directly engaged with water-based specimens under organic solvent-free conditions. As such, one promising approach is to develop enzyme biosensors based on albumin ArMs to confer water solubility and biocompatibility to metal complexes used for diagnostic purposes.

The principal goal of this paper is to demonstrate that ArMs can be adapted as enzyme biosensors to detect ethylene produced from real world biological samples. The significance of this work is that when compared to current metal-based ethylene probes, it would be one of the first to show applicability with samples directly obtained from fruits and leaves. And when compared to other ethylene detection methods (e.g. GC, electrochemical sensors), it presents a better way to analyze samples in a spatiotemporal manner.

The strategy to develop the ArM ethylene probe (**AEP**) is outlined in Fig. 1. Using the hydrophobic-binding pocket of albumin, the bound metal complex is composed of three key components: the fluorophore 7-diethylaminocoumarin (DEAC), the second generation Hoveyda–Grubbs catalyst, and the olefin-containing DABCYL quencher. In its normal state, the DABCYL quencher suppresses DEAC fluorescence via Förster resonance energy transfer (FRET) interactions. However, in the presence of ethylene gas, Hoveyda–Grubbs catalyzed cross metathesis will occur, thereby releasing the olefin-containing DABCYL quencher. As a result, detectable levels of fluorescence can be produced. To test this hypothesis, the following study investigates the ability of **AEP** to detect ethylene in biological samples that include fruits and plant leaves.

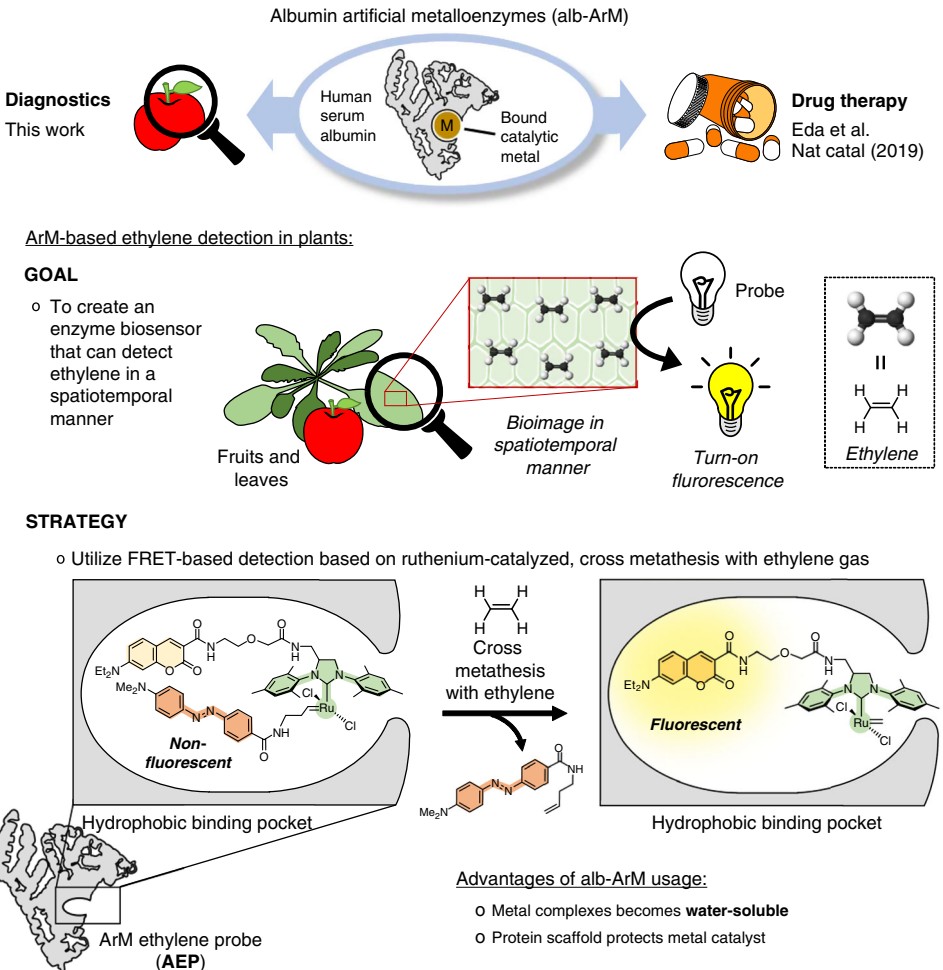

**Fig. 1 The versatility of albumin artificial metalloenzymes allows for its usage in areas such as drug therapy and diagnostics.** The goal of this study was to develop a leading example of an ArM-based enzyme biosensor, which can be used to monitor the ethylene produced in fruits and leaves via turn-on fluorescence. The strategy relies on the **AEP** probe, which uses FRET-based detection of ethylene gas through ruthenium-catalyzed, cross metathesis.

## Results

**Preparation of ethylene-sensing ArM**. The steps to prepare **AEP** are outlined in Fig. 2a. Previously in the construction of the albumin ArM (alb–Ru), a DEAC–Ru complex was utilized where a DEAC moiety was used as a ligand anchor to direct the second generation Hoveyda–Grubbs catalyst towards the hydrophobic-binding pocket of albumin. The fluorescence emitted by DEAC itself is highly sensitive to the polarity of the surrounding solvent. As shown in Fig. 2b, the quantum yield of DEAC–Ru increases by roughly 20-fold in the transition from a polar environment (10% dioxane/$H_2O$) to a non-polar environment (60% dioxane/$H_2O$). As such, alb–Ru generally exists as a fluorescent and water-soluble ArM, prepared as described in the "Methods" section.

In literature, DEAC and DABCYL have previously been identified as a FRET donor pair[42]. Therefore, the approach undertaken in this study to prepare the **AEP** probe is based on reacting alb–Ru with the olefin-containing DABCYL quencher, as described in the "Methods" section. The resulting **RuQ** complex found inside the albumin-binding pocket exhibits significantly lower levels of fluorescence due to quenching, which is evident when comparing the observed fluorescence between alb–Ru and the **AEP** probe in Fig. 2c. Furthermore, no major structural changes to the protein complexes were identified during preparation since their circular dichroism (CD) spectra overlapped perfectly (Fig. 2d).

**Ethylene-reactivity studies**. The theory behind **AEP** is that biosynthesized ethylene produced by plants will react with the bound ruthenium catalyst, thereby displacing the DABCYL quencher to activate a fluorescent signal. Given the competition with the more reactive ethylene substrate, and the added obstacle of re-entering the albumin-binding pocket under biological conditions, it is surmised that reversible reactivity of the DABCYL quencher will only have a mild influence during bioimaging.

To investigate **AEP** reactivity, a series of in vitro studies were performed. First, dose-dependent response curves were obtained for **RuQ** only (Fig. 2e). These experiments were done by injecting varying volumes of ethylene gas (50–1000 µl) into an airtight cuvette. After incubation for 1 min, the change in fluorescence was quantified. For experiments where **RuQ** was dissolved in pure organic solvent (100% THF), reactivity was shown to be comparable to contemporary examples of ruthenium-based ethylene probes[18,19]. The limit of detection (LOD) was calculated to be 34.4 µl of injected ethylene, which corresponds to about 27 ppm in air. It should be noted that since physiological levels of ethylene produced during ripening are typically <1 ppm, incubation periods for these probes would thus need to exceed times generally analogous with "real-time" detection (within 1 min). In the case where experiments were performed for **RuQ** in an aqueous environment, reactivity was instead observed to become greatly diminished due to two main factors. First, it is an expected

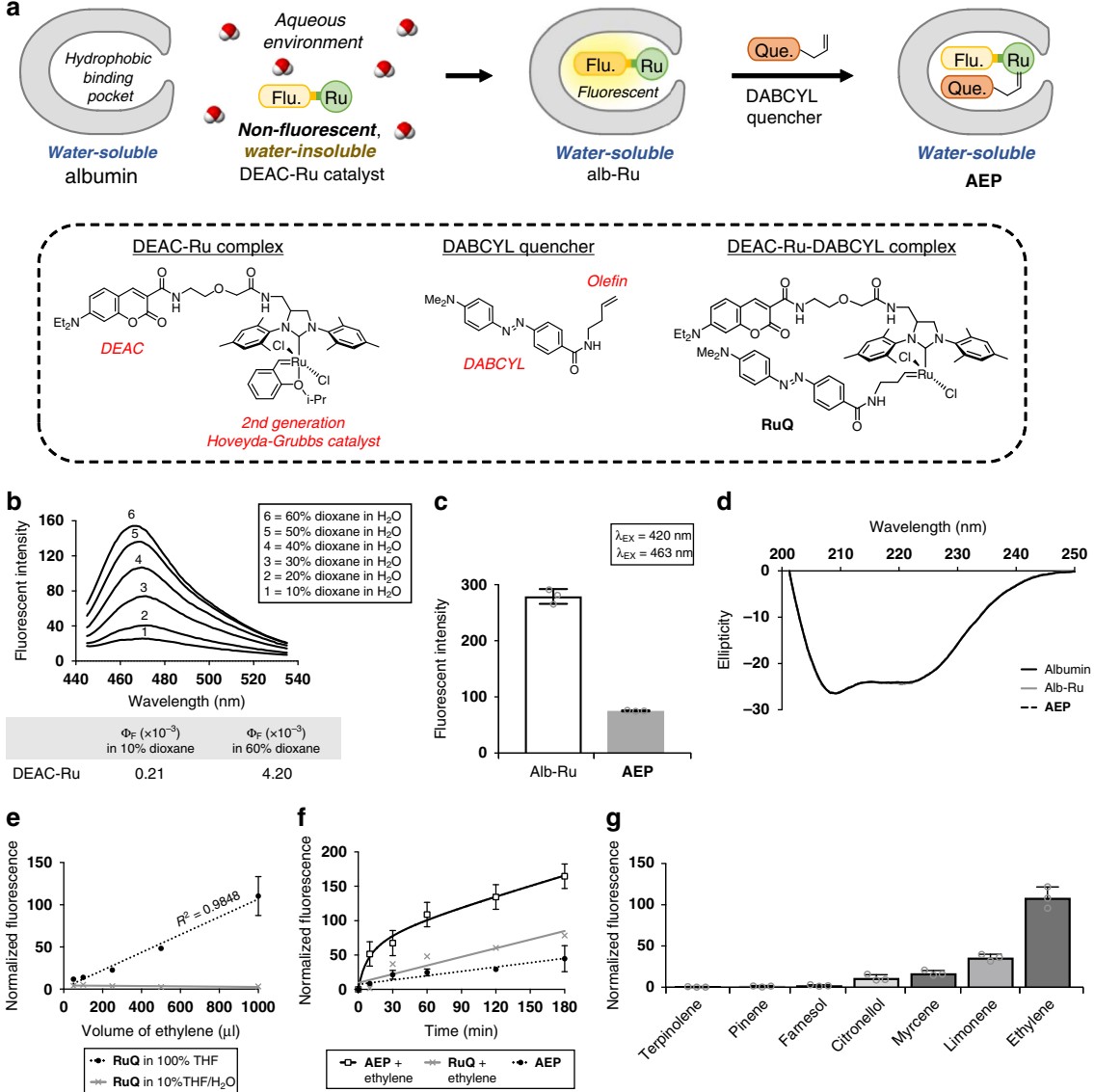

**Fig. 2 Preparation and characterization of the AEP probe. a** Depiction of the general steps taken to convert albumin into the ethylene detecting **AEP** probe. Chemical structures of the DEAC–Ru complex, DABCYL quencher, and **RuQ** are shown for reference. **b** Emission spectra and calculated quantum yields of DEAC–Ru under varying concentrations of dioxane in water. **c** Measured fluorescence of the alb–Ru complex compared to the non-fluorescent **AEP** probe. **d** Circular dichroism spectra comparison of the various protein complexes used in this study to investigate changes in structural folding. **e** Dose-dependent response curves for **RuQ** exposed to different volumes of ethylene gas under solvent conditions of either 100% THF or 10%THF/$H_2O$. **f** Time-based monitoring of **AEP** (100 μM) exposed to a theoretical concentration of 6 mM ethylene dissolved in $H_2O$. **g** Selectivity study of **AEP** (100 μM) against various types of terpenes (6 mM). In general, fluorescence studies were performed at $\lambda_{EX} = 420$ nm/$\lambda_{EM} = 463$ nm. Error is represented as s.d. of three independent experiments. Source data are provided as a Source Data file.

occurrence that olefin metathesis using the Hoveyda–Grubbs catalyst proceeds with lower reactivities under aqueous conditions. And second, a limitation of the experimental setup does not properly account for the poor solubility of ethylene in water. To highlight this, NMR studies were done using similar air-to-liquid volume ratios (Supplementary Fig. 1). With 1 ml of ethylene injected into pure solvent (100% THF-d8), the dissolved concentration was calculated to be roughly 1 mM. However, under aqueous conditions (25% or 10% THF-d8/$D_2O$), no peaks corresponding to ethylene could be detected.

In order to design an experimental setup that can ensure an adequate level of ethylene is present under aqueous conditions to test **AEP** reactivity, the following assumptions were made. According to Henry's gas law, the amount of dissolved gas in a liquid is proportional to its partial pressure above the liquid. As such, one

setup could see a closed system (with the cuvette) containing water that is purged and fitted with a balloon-containing ethylene gas. In this manner, it can be assumed that the only gas present is ethylene. Since the pressure of a balloon does not significantly deviate from 1 atm, it can be assumed that the total pressure of ethylene gas present is equivalent to its partial pressure. Using a Henry's law constant of $5.98 \times 10^{-3}$ M atm$^{-1}$, the calculated concentration of dissolved ethylene in water would thus be approximately 6 mM, or 167 ppm. To initiate time-dependent reactivity studies, **AEP** was then injected into the ethylene-filled cuvette and monitored over time. Shown in Fig. 2f, a significant time-dependent increase in fluorescence was thus observed in the **AEP**/ethylene mixture compared to controls (RuQ/ethylene and **AEP** only).

Not only does ethylene has one of the highest reactivities for olefin metathesis[43], but its smaller molecular size should allow it to

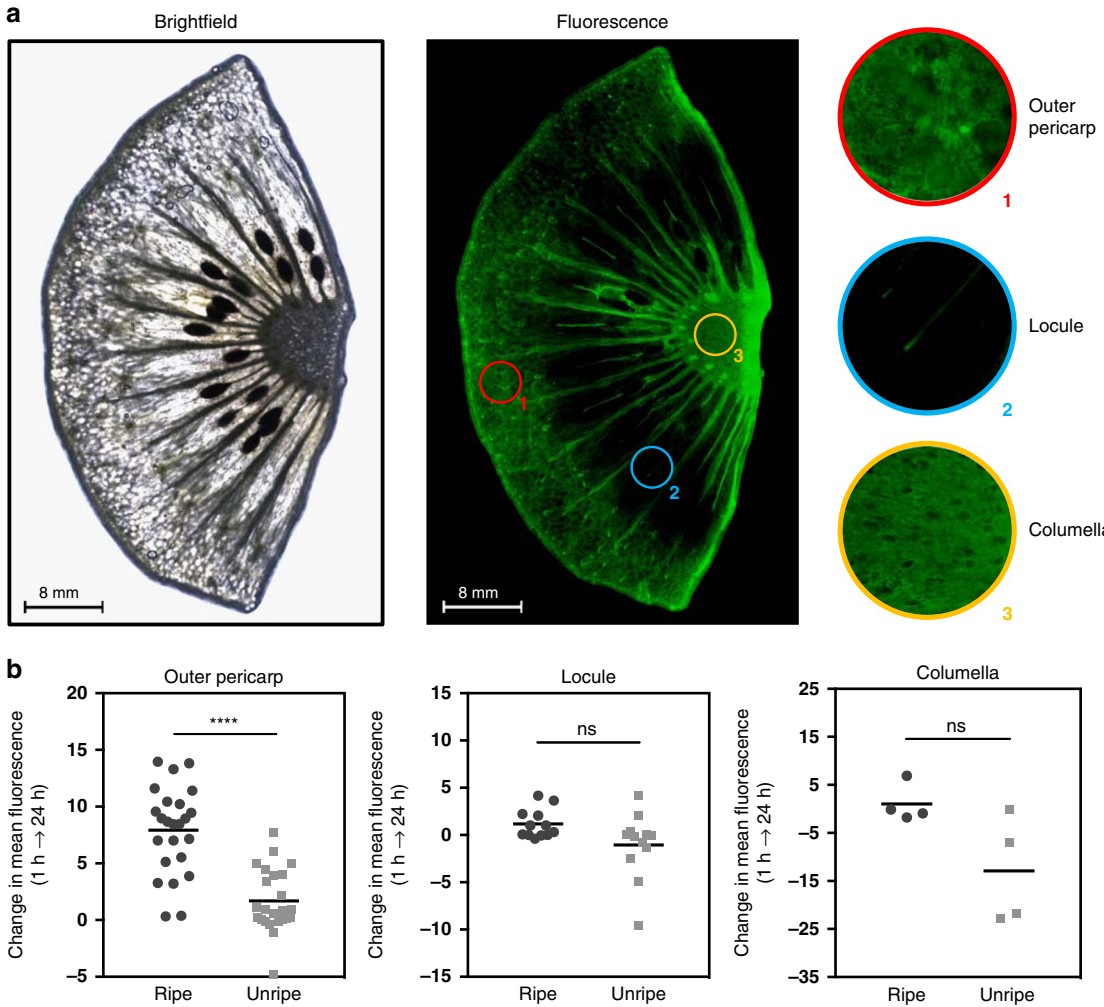

**Fig. 3 Spatial-based detection of ethylene in ripening and unripe kiwifruit. a** Fluorescent imaging of a ripening kiwi fruit, which was sectioned to display the outer pericarp, locules, and part of the columella. Multiple images were obtained at ×4 magnification and were joined together via image stitching. **b** The measured change in fluorescence of the pericarp, locule, and the columella of kiwifruit (unripe or ripe) was taken between the 1 and 24 h timepoints. As shown, a significant increase in **AEP**-detected ethylene can be observed in the pericarp during the process of ripening. For each condition (unripe or ripe), two whole fruits were used to provide four different sections for investigation. Replicates values for the outer pericarp ($n = 24$), locule ($n = 12$), and the columella ($n = 4$) were obtained. Statistical analysis was performed using a paired sample $t$-test. $*P < 0.03$, $**P < 0.002$, $***P < 0.0002$, $****P < 0.0001$, ns = not significant. Source data are provided as a Source Data file.

enter the albumin-binding pocket with better ease compared to other plant metabolites, such as unsaturated molecules known as terpenes. To determine whether there is a chance for significant cross-reactivity with **AEP**, various terpenes (e.g. pinene, limonene, terpinolene, myrcene, citronellol, and farnesol) were tested utilizing similar conditions to the previous ethylene experiments (final terpene concentration of 6 mM). Shown in Fig. 2g, the overall reactivities of the tested terpenes were found to be significantly lower than with ethylene gas. However, it should be noted that in cases containing terminal unsaturated alkenes (such as myrcene and limonene), moderate activity was observed. As such, additional controls may be required when testing samples like cannabis (myrcene-abundant) and citrus fruits (limonene-abundant).

**Spatial detection of ethylene in fruits**. Climacteric fruits typically undergo autocatalytic ethylene (system 2) production during endogenous processes, such as ripening, and also in response to exogenous stimuli, such as wounding stress and pathogen infection[44,45]. In this study, the spatial capabilities of the **AEP** probe was first investigated in response to endogenous ethylene. During ripening, studies have observed that kiwifruit exhibit an increase in

ethylene production primarily through upregulation of ACS isogenes in the outer pericarp[46]. As shown in Fig. 3a and Supplementary Figs. 2–9, imaging experiments in kiwifruits using **AEP** focused on observing changes in ethylene expression of various organelles (outer pericarp, locule, and columella) at different stages of development (unripe vs. ripening). Summarized in Fig. 3b, a significant increase in fluorescence was detected in the outer pericarp from the transition of unripe to ripe fruit. On the other hand, fluorescent changes in the locule and columella of the kiwifruits did not occur at significant levels. As a note, it was observed that the tissues between individual locules generally exhibited an increase in fluorescence. Since literature has shown that kiwifruit seeds or surrounding tissues may play a role in ethylene biosynthesis (parthenocarpic kiwifruit are shown to have decreased levels of ethylene production due to suppressed expression of ACS[47]), a more thorough analysis may be of interest for future studies. Overall, with these results, the use of **AEP** presents itself as a promising tool to monitor the spatial expression of ethylene for fruits during ripening and development.

The ethylene biosynthetic pathway in plants (Fig. 4a) is mainly composed of three enzymes that convert methionine into

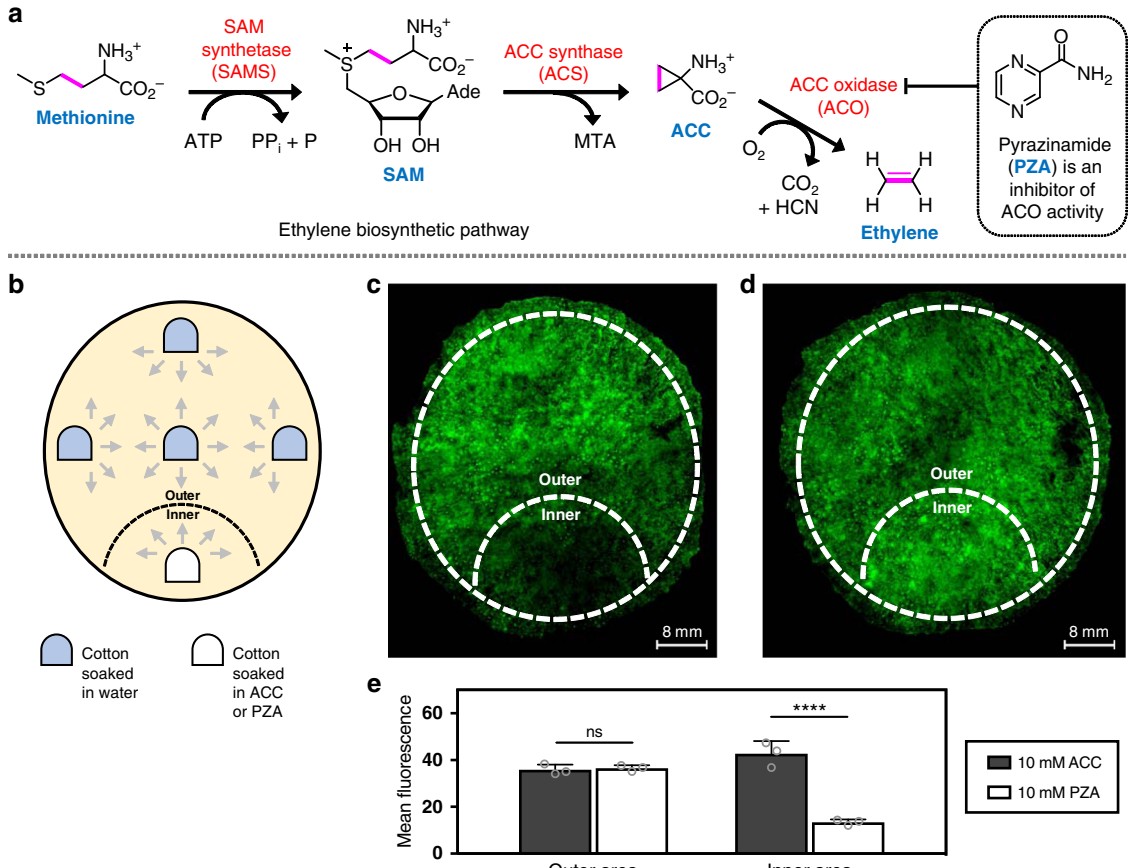

**Fig. 4 Spatial-based detection of ethylene in sections of Asian pears induced with either an ethylene-stimulating or ethylene-inhibiting additive. a** Scheme of the ethylene biosynthetic pathway in plants. Addition of ACC, the metabolite directly preceding ethylene biosynthesis, will stimulate ethylene production. Addition of PZA, a known inhibitor of ACC oxidase, will decrease ethylene production. **b** Cotton balls soaked in specific solutions (water, ACC, or PZA) are gently applied to sections of Asian pears. To minimize diffusion, five cotton balls are applied with equal spacing to the surface of pear sections. Four of them contain water, while one of them has been soaked in either 10 mM ACC or 10 mM PZA. **c** Fluorescent imaging of an Asian pear section where the inner area has been exposed to 10 mM PZA. **d** Fluorescent imaging of an Asian pear section where the inner area has been exposed to 10 mM ACC. **e** Summary of measured fluorescence between the outer and inner areas of Asian pear sections that have been supplemented with either ACC or PZA. Multiple images were obtained at ×4 magnification and were joined together via image stitching. Statistical analysis was performed using a one-way ANOVA with Tukey's multiple comparisons test. *$P < 0.03$, **$P < 0.002$, ***$P < 0.0002$, ****$P < 0.0001$, ns = not significant. Error is represented as s.d. of three independent experiments. Source data are provided as a Source Data file.

ethylene: SAM synthetase (SAMS), ACC synthase (ACS), and ACC oxidase (ACO)[48]. Although ACS is often regarded as the rate-limiting step of ethylene biosynthesis[49], recent research has indicated that ACO may also be rate limiting during post-climacteric ripening of tomatoes (and likely other climacteric fruit)[50,51]. Of relative importance to this study, pyrazinamide (PZA) was recently found to be an inhibitor of ethylene biosynthesis through inhibition of ACO[52]. With consideration of this pathway, this study generally relied on using ACC as a stimulant of ethylene production, and PZA as a repressor of ethylene production.

In another investigation, the ability of **AEP** to respond to changes in ethylene biosynthesis due to external stimuli was carried out. Cotton balls soaked in either a solution of ACC or PZA were topically applied to sections of fruit flesh (hypanthium) from Asian pears. In order to ensure uniform hydration of the sample, water-infused cotton balls were placed in an equally spaced arrangement depicted in Fig. 4b. Following a 5 h incubation period, cotton balls were removed and the samples were then placed on plates preloaded with **AEP**. Fluorescence microscopy-imaging results obtained after 18 h are shown in Fig. 4c, d and Supplementary Figs. 10 and 11. In this study, the inner area is defined as the tissue

roughly within a 15 mm radius from the origin of stimulant (ACC or PZA) application. On the other hand, the outer area is the tissue area exposed to water (as a control). As summarized in Fig. 4e, results clearly show differences in inner area fluorescence compared to outer area fluorescence. For example, exposure to ACC was shown to moderately increase inner area fluorescence by about 20%. Alternatively, exposure to PZA gave the opposite effect by greatly decreasing inner area fluorescence by up to 60%. Although promising, it should be noted that there are major experimental challenges associated with studying spatial changes in ethylene biosynthesis due to external stimuli; the main culprit being the issue of diffusion. Without a strong external inducer, diffusion of the stimulant greatly decreases spatial resolution. This is especially problematic for samples with smaller surface areas, as similar experiments could not be reproduced with smaller plant leaves.

**Temporal detection of ethylene in fruits.** To investigate the temporal detection capabilities of the **AEP** probe across a variety of different fruits, the general protocol outlined in Fig. 5a was followed. To begin, small sections of fruit measuring roughly 10 mm × 8 mm in size were prepared. These samples were then incubated with varying concentrations of ACC or PZA for 3 h at

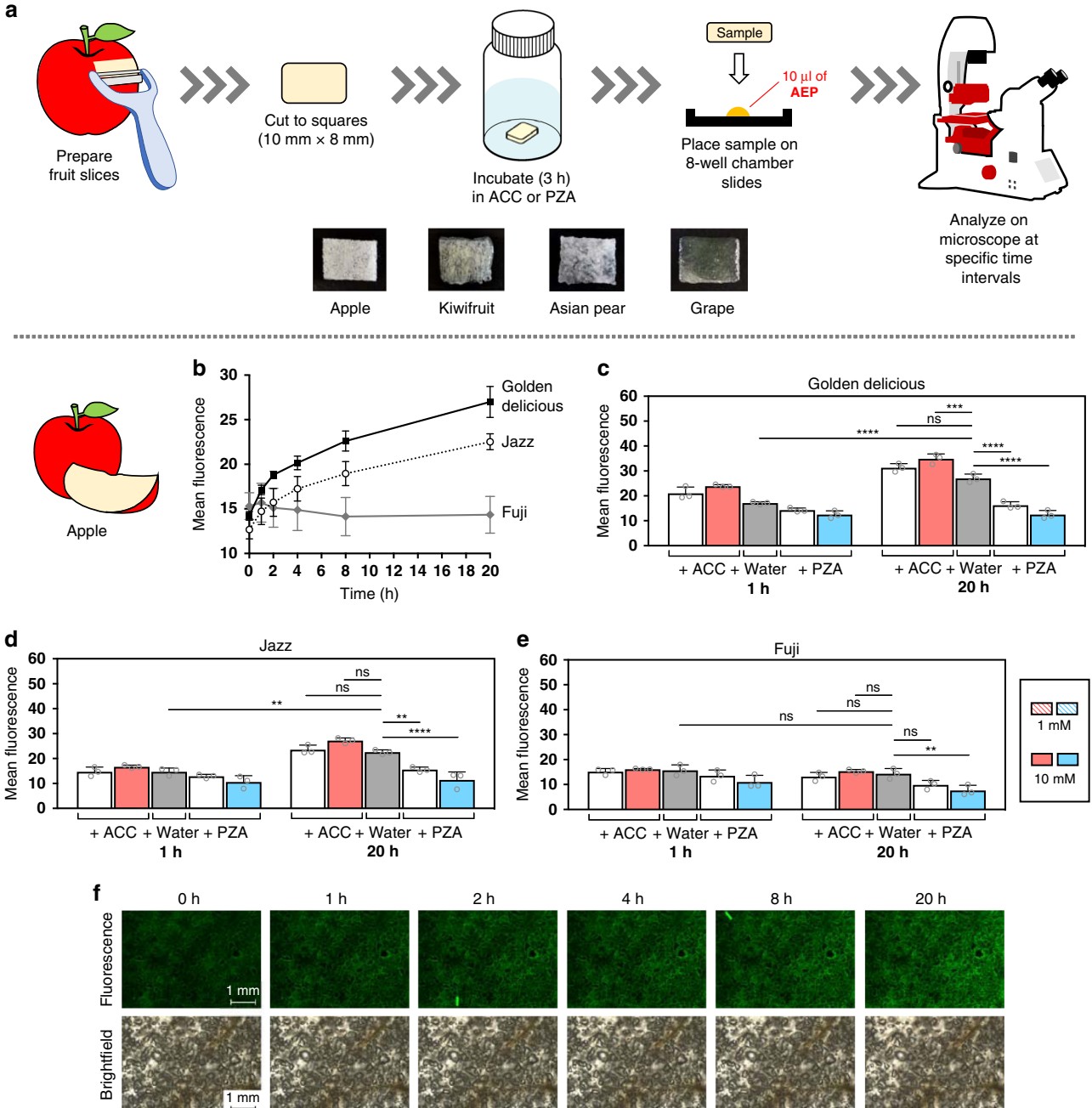

**Fig. 5 Time-dependent detection of ethylene in apples. a** In general, samples were prepared from slices of fruit flesh cut into ~10 mm × 8 mm segments. Following a 3 h incubation either in solutions of ACC, PZA, or water (as a control), samples were then removed and placed onto chamber slides preloaded with **AEP** probe (10 µl of a 400 µM solution). Using an inverted microscope, samples were monitored at specific time intervals. To examine the difference in ethylene production across different apple cultivars, fluorescence was monitored over time for Golden Delicious, Jazz, and Fuji apples **b**. A summary and statistical comparison between the 1 and 20 h timepoints for each apple cultivar are also shown **c–e**, as well as samples images of Golden Delicious apples (water-control) over various timepoints **f**. Statistical analysis was performed using a one-way ANOVA with Tukey's multiple comparisons test. *$P < 0.03$, **$P < 0.002$, ***$P < 0.0002$, ****$P < 0.0001$, ns = not significant. Error is represented as s.d. of three independent experiments. Source data are provided as a Source Data file.

room temperature, before being placed onto chamber slides preloaded with **AEP**. Fluorescence analysis was then carried out at specific time intervals.

In literature, internal ethylene levels have been shown to vary among different types of apple cultivars[53–55]. For instance, Fuji apples are known to produce lower levels of ethylene gas compared to other cultivars[55]. To test the capacity of **AEP** to detect these differences, time-based studies were carried out using apple cultivars that included Golden Delicious, Jazz, and Fuji

apples (Fig. 5b–f and Supplementary Figs. 12–26). As summarized in Fig. 5b, data clearly shows that Fuji apples produce lower levels of ethylene compared to other cultivars over time. Furthermore, this trend was also observed during supplementation studies using ACC or PZA (Fig. 5c–e).

Additionally, other fruits were also examined in this study using the **AEP** probe, which included kiwifruit, Asian pears, and Muscat grapes (Fig. 6 and Supplementary Figs. 27–41). For climacteric fruits like kiwifruit and Asian pears[56,57], both

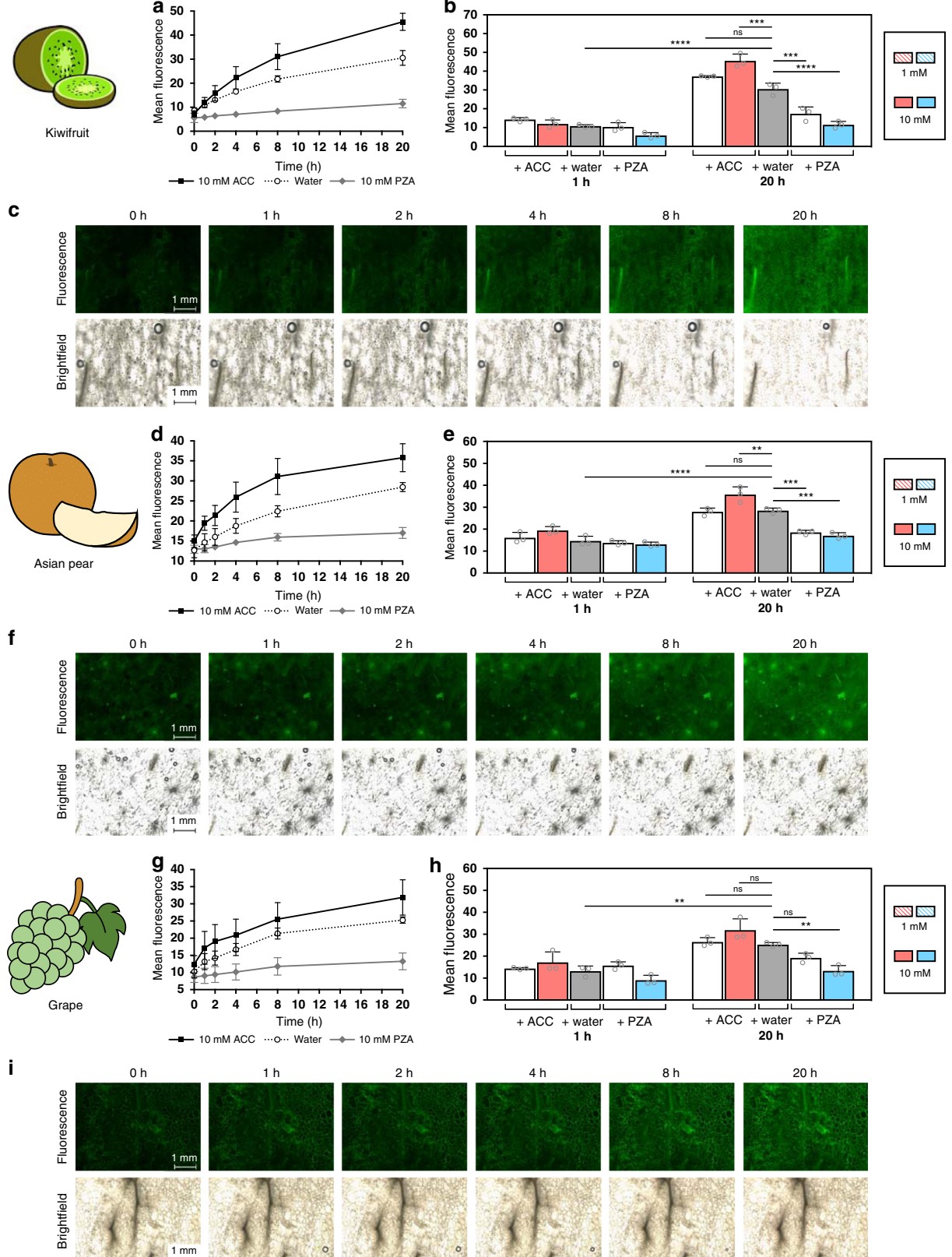

**Fig. 6 Time-dependent detection of ethylene in other ethylene-producing fruits.** Fluorescence was monitored over time under various conditions (ACC or PZA) for kiwifruit **a**, Asian pear **d**, and Muscat grapes **g**. A summary and statistical comparison between the 1 and 20 h timepoints for each fruit are shown **b**, **e**, **h**, as well as samples images (water-control) over various timepoints **c**, **f**, **i**. Statistical analysis was performed using a one-way ANOVA with Tukey's multiple comparisons test. *$P < 0.03$, **$P < 0.002$, ***$P < 0.0002$, ****$P < 0.0001$, ns = not significant. Error is represented as s.d. of three independent experiments. Source data are provided as a Source Data file.

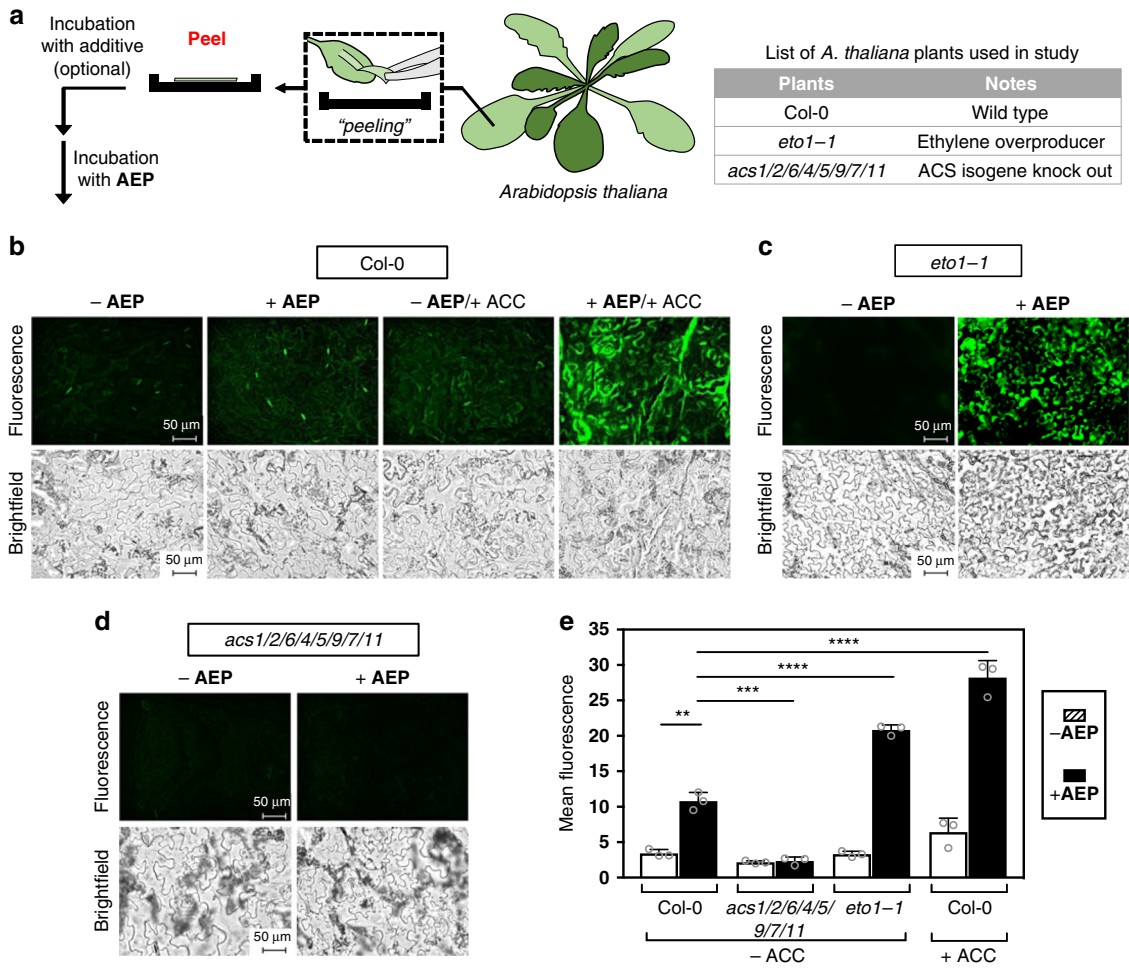

**Fig. 7 Detection of ethylene produced in *A. thaliana* plants. a** Depiction of sample preparations to obtain epidermal peels from *A. thaliana* and the list of plants used in this experiment. Fluorescence and brightfield imaging (×40 magnification) of epidermal peels applied with **AEP** (100 μM) for wild-type Col-0 **b**, ethylene overproducer *eto1-1* **c**, and the ethylene biosynthesis defected *acs1/2/6/4/5/9/7/11* **d**. Additionally, supplementation studies were also performed using the ethylene biosynthetic precursor ACC (1 mM), which was only applied to wild-type Col-0. **e** Summary of the measured fluorescence between the various *A. thaliana* plants applied with and without **AEP**, as well as the effects of ACC supplementation. Statistical analysis was performed using a one-way ANOVA with Tukey's multiple comparisons test. *$P < 0.03$, **$P < 0.002$, ***$P < 0.0002$, ****$P < 0.0001$, ns = not significant. Error is represented as s.d. of three independent experiments. Source data are provided as a Source Data file.

displayed time-dependent ethylene production as expected. Interestingly, samples from Muscat grapes were also tested and produced ethylene in a time-dependent manner. Although generally classified as non-climacteric, studies have shown grapes have the capacity to produce ethylene gas; reaching its peak during grape maturation[58]. Furthermore, detectable levels of ACS and ACO gene expression can be found long after development and ripening[59]. As a step to further check **AEP** sensitivity, negative controls were carried out with ethylene non-producing vegetables like carrots and red bell peppers (Supplementary Figs. 42–52). As expected, these cases produced non-significant levels of ethylene over similar time periods.

**Ethylene detection in plants**. To investigate the efficacy of **AEP** detection in plants, the small flowering plant *Arabidopsis thaliana* (Brassicaceae family) was chosen as the model organism. Focusing specifically on imaging of clear epidermal peels, comparative studies were carried out using a wide range of small molecules and plant mutants (Fig. 7 and Supplementary Figs. 53 and 54).

For this study, the Col-0 ecotype of *A. thaliana* served as the model wild-type plant. Also used were various mutants; *acs1/2/6/*

*4/5/9/7/11* and *eto1-1*. The mutant *acs1/2/6/4/5/9/7/11*, which contains knock outs of eight functional ACS genes, was chosen because it exhibits dramatically reduced levels of ethylene that persist even during pathogen invasion[60,61]. This is due to the key role that ACS plays in the ethylene biosynthetic pathway (Fig. 4a). Another of the chosen mutants is *eto1-1*, which is characterized by ethylene overproduction[62]. In this case, the *eto1-1* mutation results in the stabilization of type II ACS, thereby preventing proteasomal degradation of ACS[63].

Shown in Fig. 7b are the images obtained for Col-0 incubated with and without **AEP** at room temperature. As quantified in Fig. 7e, a significant increase in fluorescence was observed, thereby validating the use of **AEP** for ethylene detection in *A. thaliana* epidermal peels. As a positive control, ethylene production was contrasted with Col-0 exogenously stimulated by ACC incubation. As expected, **AEP**-based ethylene detection showed a significantly higher increase in detected fluorescence compared to the wild type Col-0.

To further evaluate the sensitivity of the **AEP** probe to detect endogenous differences in ethylene production, comparative studies were then carried out using mutants acting as either a positive ethylene control (*eto1-1*) or a negative ethylene control

(*acs1/2/6/4/5/9/7/11*). Shown in Fig. 7c, imaging for the ethylene overproducer, *eto1-1*, showed a significant increase in fluorescence compared to the wild type Col-0, as expected. In contrast, with the *acs1/2/6/4/5/9/7/11* mutant that naturally lacks the ability to produce ethylene, there was no discernible fluorescent change with **AEP** addition (Fig. 7d). When compared with the wild-type Col-0, the *acs* knockout plant thus exhibited significantly lower levels of fluorescence.

In the next part of this study, **AEP** probe sensitivity was tested in response to environmental stress factors, namely the immunogenic response to pathogen invasion in *A. thaliana*. Focus will be given to changes in ethylene production caused by two separate plant immunity pathways: PAMP-triggered immunity (PTI) and effector-triggered immunity (ETI).

**PTI-based ethylene detection.** Pathogenic bacteria attempting to invade plant cells initially must contend with the first tier of plant's defense mechanism—PTI. Depicted in Fig. 8a, this system is triggered by the recognition of pathogen-derived molecules known as pathogen-associated molecular pattern (PAMP) through pattern recognition receptors (PRRs) on the plasma membrane. Once triggered, a signaling cascade then commences to upregulate genes associated with the defense response. Among the upregulated genes, studies have shown that ethylene biosynthesis plays an integral role in PTI[64], possibly due to its regulatory role for innate immune receptors[65].

Two well-studied PAMPs used in this study are the flg22 and elf18 bacterial PAMPs. Flagellin is a globular protein found in the filament that composes the bacterial flagellum. On the conserved

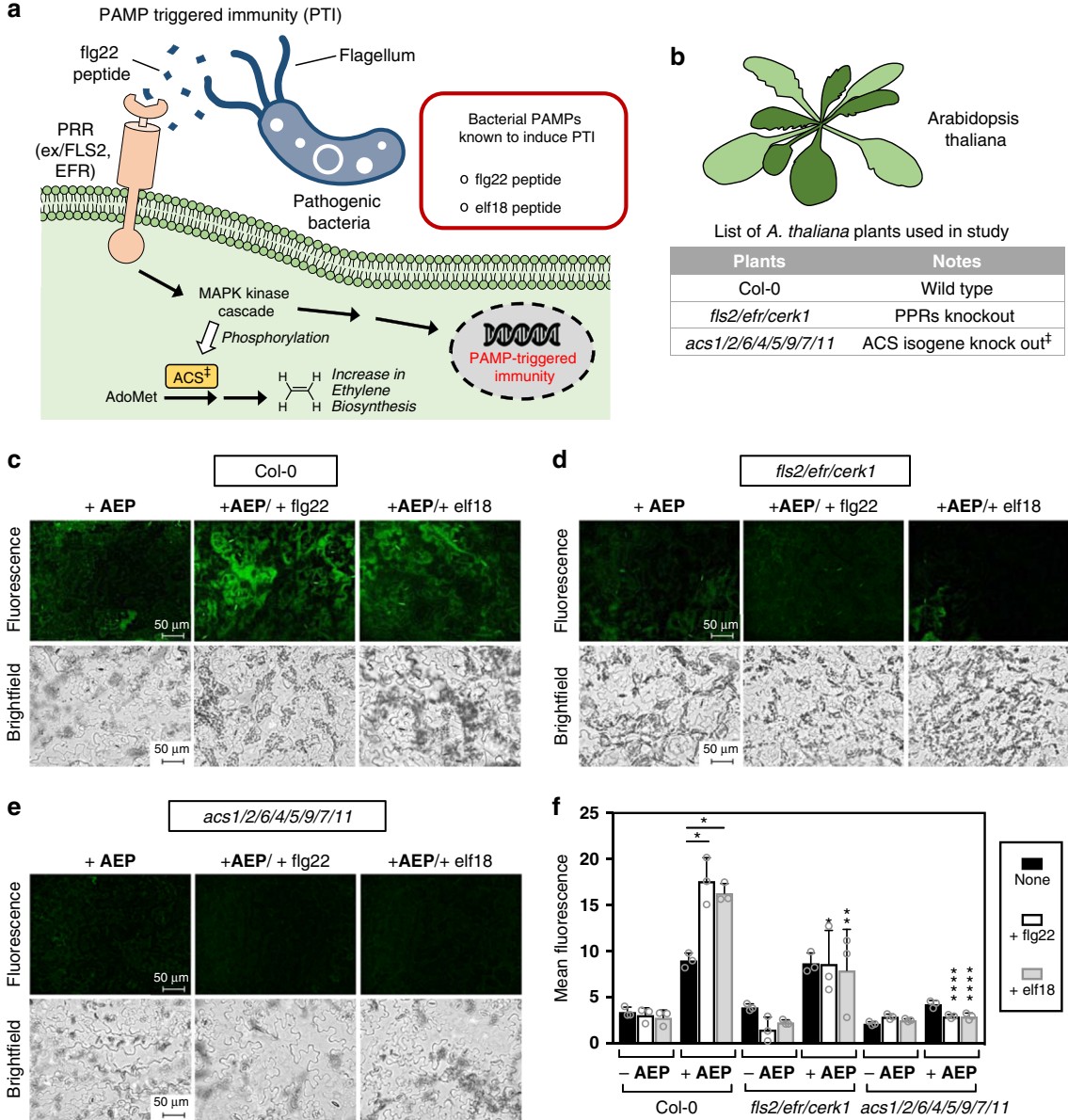

**Fig. 8 Detection of ethylene produced as a result of PAMP-triggered immunity in *A. thaliana* plants. a** Depiction of the general pathway that leads to PTI response. **b** List of *A. thaliana* plants used in this experiment. Fluorescence and brightfield imaging (×40 magnification) of epidermal peels applied with **AEP** (100 μM) for wild-type Col-0 **c**, PRR-deficient *fls2/efr/cerk1* **d**, and the ethylene biosynthesis defected *acs1/2/6/4/5/9/7/11* **e**. Additionally, supplementation studies were performed using the PAMPs, flg22, and elf18 (4.8 μM), which were applied to all plants. **f** Summary of the measured fluorescence between the various conditions studied. Statistical analysis was performed using a one-way ANOVA with Tukey's multiple comparisons test. If not indicated, comparisons were made between the mutants and wild-type under the same treatment conditions. *$P < 0.03$, **$P < 0.002$, ***$P < 0.0002$, ****$P < 0.0001$, ns = not significant. Error is represented as s.d. of three independent experiments. Source data are provided as a Source Data file.

N-terminus of flagellin, a 22-amino acid sequence known as flg22 has been found to activate plant defense mechanisms[66,67]. It does this by first binding to the PRR known as FLS2, which then leads to the activation of MAPK-signaling cascades and the production of reactive oxygen species (ROS) to confer pathogen resistance[68]. For the bacterial protein EF-Tu (elongation factor thermo unstable), an 18-amino acid sequence, termed elf18, is known to be important for post-invasive immunity and capable of eliciting PTI[69]. In this case, perception instead acts through the PRR known as EFR (EF-TU receptor), followed by downstream MAPK signaling and ROS production[70].

In this study, epidermal peels from *A. thaliana* Col-0 were first incubated for 6 h with either PAMP (flg22 or elf18) or a blank control. Incubation with **AEP** was then carried out, followed by imaging and quantification (Fig. 8c–e and Supplementary Figs. 55–60). As shown in Fig. 8c, significant increases in fluorescence could be detected with tissues incubated with either PAMP compared to the blank control. This data correlates well with literature reports that have shown consistently high levels of ethylene gas generated after flg22 or elf18 exposure[61].

As a further investigation, control studies were also carried out using the mutant plants *fls2/efr/cerk1* and *acs1/2/6/4/5/9/7/11*. For the *A. thaliana* mutant known as *fls2/efr/cerk1*, the two PAMP-sensing PRR genes FLS2 and EFR are knocked out, leaving these plants unable to sense and respond to the presence of extracellular PAMP. With the images shown in Fig. 8d, fluorescence levels in the mutants incubated with either flg22 or elf18 falls to levels on par with the blank controls, thereby representing a significant decrease compared to the corresponding wild type Col-0. With the *acs1/2/6/4/5/9/7/11* mutant, no discernible difference in fluorescence with epidermal peels incubated with or without PAMPs was found (Fig. 8e). Consistent with the fact that this mutant is unable to biosynthesize ethylene[60,61], observed fluorescence was again significantly reduced compared to the corresponding wild type Col-0.

**ETI-based ethylene detection.** In case the first defense system (PTI) is defeated, innate resistance mechanisms in plants initiate a second defense system known as ETI. As depicted in Fig. 9a, the basis behind this immune response relies on the fact that bacterial pathogens are known to infect eukaryotic cells using an appendage to secrete toxins or effectors into the host cells by type III secretion system (or injectisome). These secreted effectors have the ability to exert a number of effects that promote pathogen virulence. Examples of well-studied effectors are AvrRpm1 and AvrRpt2, which can be produced by *Pseudomonas syringae* pv. *tomato* (Pst) to interfere with the PTI-regulating membrane protein RPM1-interacting protein 4 (RIN4)[71]. To combat this, resistant plants have resistance (R) proteins that recognize directly or indirectly some of these effectors (known as avirulent effectors), leading to a signaling cascade that upregulates genes associated with the defense response. For AvrRpm1, once associated with RIN4, phosphorylation of RIN4 is induced, followed by activation of the R protein known as RPM1[72]. In the case of AvrRpt2, degradation of RIN4 occurs instead. Since the R-protein RPS2 also associates with RIN4, its degradation essentially frees RPS2 to interact and be activated by AvrRpt2[73,74].

In this study, ETI-induced ethylene production was monitored in *A. thaliana* via exposure to two different bacterial strains known to produce either the AvrRpm1 or AvrRpt2 avirulence proteins. To carry out this study, inoculation of epidermal peels with *Pst* AvrRpm1, *Pst* AvrRpt2, or a 10 mM MgCl₂ solution was first done 12 h before detection. Incubation with **AEP** was then carried out, followed by imaging and quantification (Fig. 9c–e and Supplementary Figs. 61–66). As shown in Fig. 9c, the Col-0

epidermal peels that were exposed to *Pseudomonas* producing either AvrRpm1 or AvrRpt2 showed significantly increased levels of fluorescence compared to the MgCl₂ control.

Again, as a comparative study, *A. thaliana* mutants were also tested as controls to highlight the robustness of the **AEP** probe. One of the mutants used was *rpm1rps2*, which has both the R proteins, RPM1 and RPS2, knocked out. In this regard, the *rpm1rps2* mutant is unable to properly activate ETI even after infection with *Pseudomonas* bacteria carrying AvrRpm1 or AvrRpt2. Shown in Fig. 9d, imaging results clearly show that fluorescence levels in the mutant plant exposed to either AvrRpm1 or AvrRpt2 did not have elevated levels of ethylene production. Instead, fluorescence levels remain on par with the MgCl₂ controls. In comparison to the corresponding wild type Col-0, this signifies a dramatic reduction in detected ethylene. Finally, in the case with the *acs1/2/6/4/5/9/7/11* mutant, no discernible differences in fluorescence was found with exposure to AvrRpm1 or AvrRpt2 (Fig. 9e), which agrees well with literature[61].

**Limitations.** Through testing numerous different samples in this study, some limitations of the **AEP** probe were gradually noted. First, one key issue is absorption. Given that the exterior scaffold of **AEP** is human serum albumin (~66 kDa protein), the probe is unlikely to passively penetrate across the cell membrane and into plant cells. It is assumed that the **AEP** probe exists extracellularly, rather than intracellularly. Due to this, the texture of a sample becomes a limiting factor. When used with absorbent samples, such as the flesh of fruits, the **AEP** probe was observed to sufficiently remain on sample surfaces. However, when testing with samples of waxy nature (i.e. plant cuticles), it was noted that the **AEP** probe tends to get pushed off over time.

Another issue is related to the aspect of "real-time" detection. Due to its poor water solubility, ethylene rapidly transitions from solution into the atmosphere. As such, any (water-based) bioimaging reagent used to directly measure this phytohormone in "real-time" must react under a very short period of time. At its current state, however, **AEP** probe reactivity is not fast enough to measure "real-time". Conceivable approaches to improve upon this area could see changes made to (1) increase ruthenium reactivity by altering the metal–ligand complex, (2) change the protein scaffold for one with a shallow-binding pocket to allow easier access, or (3) adopt the use of a stronger FRET donor–acceptor pair.

**Discussion**

Overall, the work conducted in this study should have a significant impact for the advancement of two separate research fields, namely in the areas of enzyme biosensor development and also research focused on ethylene detection in plants.

Given the biocompatible nature of enzyme biosensors, their usage for biomedical applications not only has implications from a scientific perspective, but also from a commercial standpoint. Currently, numerous examples of enzyme biosensors are applied commercially, such as arylacylamidase, β-hydroxybutyrate dehydrogenase, salicylate hydroxylase, and uricase, which are used for the serum determination of metabolites like acetaminophen, ketone bodies, salicylate, and uric acid, respectively. However, since enzyme biosensors are mainly derived from naturally occurring enzymes, recent progress in this field has largely appeared to stall. In parallel, a rapid rise of interest is now being directed towards the use of biosensors composed of abiotic metal complexes, where the bioorthogonality of certain transition metals can help to target metabolites typically found in difficult-to-access chemical spaces.

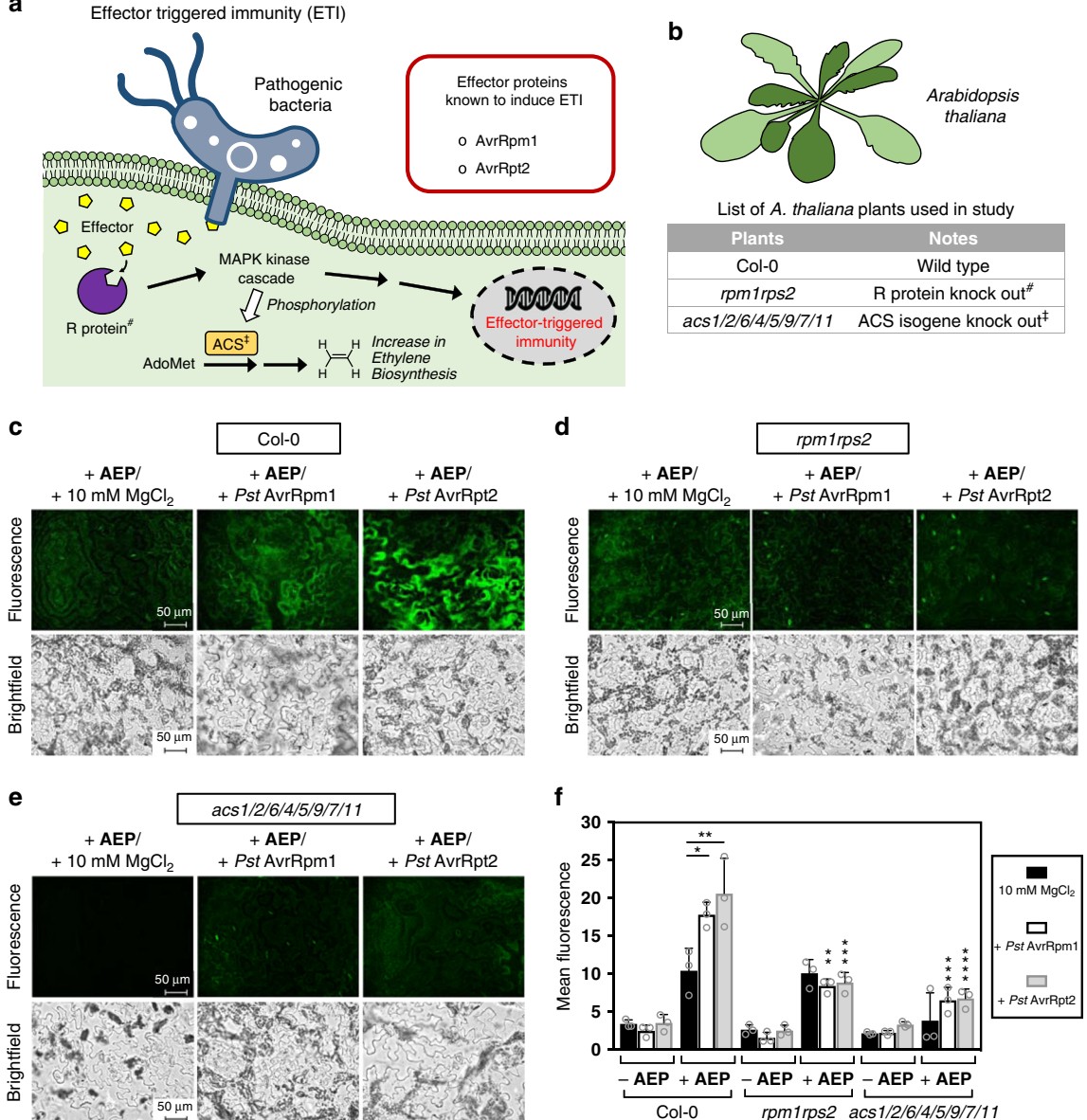

**Fig. 9 Detection of ethylene produced as a result of effector-triggered immunity in *A. thaliana* plants. a** Depiction of the general pathway that leads to ETI response. **b** List of *A. thaliana* plants used in this experiment. Fluorescence and brightfield imaging (×40 magnification) of epidermal peels applied with **AEP** (100 μM) for wild-type Col-0 **c**, R-protein-deficient *rpm1rps2* **d**, and the ethylene biosynthesis-defected *acs1/2/6/4/5/9/7/11* **e**. Additionally, supplementation studies were also performed using *Pst* AvrRpm1 or *Pst* AvrRpt2 (bacterial solution OD$_{600}$ = 0.02), which was applied to all plants. **f** Summary of the measured fluorescence between the various conditions studied. Statistical analysis was performed using a one-way ANOVA with Tukey's multiple comparisons test. If not indicated, comparisons were made between the mutants and wild-type under the same treatment conditions. *$P < 0.03$, **$P < 0.002$, ***$P < 0.0002$, ****$P < 0.0001$, ns = not significant. Error is represented as s.d. of three independent experiments. Source data are provided as a Source Data file.

With the advent of ArMs, a highly unique opportunity exists for the development of "artificial" enzyme biosensors, where metabolite detection can be driven via abiotic metal catalytic reactions. To the best of our knowledge, this study represents a leading example of an ArM being developed into an enzyme biosensor that has shown robust application with real world biological samples, such as fruits and leaves. Given its ease of preparation, there is the added benefit that this work can help pioneer the field of ArM biosensors, where different transition metals (e.g. palladium, rhodium, ruthenium, etc.) can be exploited for the chemoselective detection and quantification of other important biological metabolites.

Another significant aspect of this work is related to the development of the ethylene-sensing **AEP** probe, which has proven to be capable of detecting changes (both induced endogenously and exogenously) in biological concentrations of ethylene in various fruits and *A. thaliana* plants. Given the physiological importance of ethylene for plants, numerous studies have been conducted to understand ethylene-dependent pathways. However, one limitation of current detection methods (e.g. gas chromatography) is that typically only the average ethylene gas released into the surrounding environment can be measured. Without a proper bioimaging technique that can spatially detect ethylene on a cellular level, it is presently a challenge for scientists

to determine the location and source of ethylene production. Despite some restrictions, this study has developed an ethylene-sensing bioimaging technique shown to be capable of measuring ethylene in a spatiotemporal manner for cross-sections of various fruits and leaves.

## Methods

**General materials**. Reagents and buffer components were purchased from Sigma-Aldrich, Fisher Scientific, Alfa Aesar, TCI, or Wako Chemicals without further purification. Human serum albumin (lyophilized powder, essentially fatty acid free) was purchased from Sigma-Aldrich (Prod# A1887). All experiments dealing with air-sensitive and moisture-sensitive compounds were conducted under an atmosphere of dry nitrogen. Anhydrous solvents were used as received, which include THF (anhydrous; Fujifilm Wako Pure Chemical), dichloromethane (anhydrous; Fijifilm Wako Pure Chemical) and benzene (anhydrous; Fujifilm Wako Pure Chemical). TLC analyses (F-254) were performed with 60 Å silica gel from Merck. $^1$H and $^{13}$C NMR spectra were measured on either a JEOL JNM-AL300 (300 MHz), JNM-ECZ400R/S1 (400 MHz), or JNM-ECA500 (500 MHz) instrument with the solvent peaks as internal standards: δH 0.00 (tetramethylsilane) and δC 77.0 for CDCl$_3$. For chemical synthesis, high-resolution mass spectra (HRMS) were obtained on a Bruker MicroTOF-QIII spectrometer® by electron spray ionization time-of-flight (ESI-TOF-MS). Ultrapure water used for all experiments was obtained from a Milli-Q Advantage® A10 Water Purification System sold by Merck Millipore (Burlington, USA). In addition, Amicon® Ultra Centrifugal Filters (10 and 30 kDa) and Durapore PVDF 0.45 μm® filters were also purchased from Merck Millipore (Burlington, USA). All fruit and vegetable samples were purchased from a local grocery store. Previously published *A. thaliana* mutants were: *rpm1rps2* (*rpm1-3/rps2-101C*)[72], *eto1-1*[62], *acs1/2/6/4/5/9/7/11*[60], and *fls2/efr/cerk1*[66,70]. The following bacterial strains were used in this study: *P. syringae* pv. tomato DC3000 transformed with pVSP61-AvrRpt2 (*Pst* AvrRpt2) and pVSP61-AvrRpm1 (*Pst* AvrRpm1)[75]. Bacteria were propagated on LB media containing selection antibiotics and suspended with a 10 mM MgCl$_2$ solution at a concentration of $OD_{600} = 1.0$.

**Preparation of AEP**. As previously described[34], alb–Ru was prepared first by mixing 30 μM of human serum albumin (50 nmol, 167 μl from 300 μM stock solution in H$_2$O) and 37 μM of DEAC-Ru (62 nmol, 167 μl from 370 μM stock solution in dioxane). The total reaction volume was adjusted to 1670 μl of 10% dioxane in PBS buffer pH 7.4. Following initiation by HSA addition, reaction mixtures were mildly mixed and incubated at 37 °C for 1 h. The reaction mixture was then concentrated and washed with PBS buffer (3×) using Amicon® ultra centrifugal filters (30 kDa). The concentrated alb–Ru solution was then diluted in H$_2$O to 50 μl to obtain a 1 mM stock solution. To prepare the **AEP** solution, a mixture of 100 μM of alb–Ru (50 nmol, 50 μl from 1 mM stock solution in H$_2$O) and 500 μM DABCYL quencher (250 nmol, 450 μl from 555 μM stock solution in 1:8 DMSO in H$_2$O) was made. Solutions were mildly mixed and incubated at 37 °C for 5 min. The reaction mixture was then concentrated and washed with H$_2$O (3×) using Amicon® ultra centrifugal filters (30 kDa). The concentrated **AEP** solution was then diluted in H$_2$O to 500 μl to obtain a 100 μM stock solution. Following preparation, **AEP** was characterized via fluorescence spectroscopy and CD.

**Protein complex fluorescence**. For the protein complexes used in this study (alb–Ru, **AEP**), fluorescence was determined via spectrofluorometric analysis using a JASCO FP-6500 Spectrofluorometer equipped with a JASCO FMP-963 microplate reader. For sample preparations, both alb–Ru and **AEP** were adjusted to a concentration of 10 μM in H$_2$O. Samples were then aliquoted (100 μl) into 96-well microtiter plates, and monitored at $\lambda_{EX} = 420$ nm/$\lambda_{EM} = 463$ nm. All measurements were taken in triplicate from distinct samples.

**CD experiments**. In order to identify major structural changes to the various protein complexes used in this study (alb–Ru, **AEP**), CD analysis was done. CD spectra were measured on a J-1500 CD spectrometer (JASCO, Tokyo, Japan) using a 0.1 cm cell. A solution of 10% dioxane in water was used as a blank and automatically subtracted from the samples during scanning. Data was recorded from 200 to 250 nm with a scan speed of 100 nm/min. The concentration of each protein complex was maintained at 2.3 μM.

**Ethylene concentration by NMR analysis**. As an internal standard, a 100 mM stock solution of 1,3-propanediol in THF-d8 was first prepared. For the 100% THF condition, an NMR tube was filled with 60 μl of the internal standard solution and 540 μl of THF-d8. For the 25% THF/H$_2$O condition, an NMR tube was filled with 60 μl of the internal standard solution, 90 μl of THF-d8, and 450 μl of D$_2$O. For the 10% THF/H$_2$O condition, an NMR tube was filled with 60 μl of the internal standard solution and 540 μl of D$_2$O. Tubes were gastight sealed, before 1 ml of ethylene gas was injected into each NMR tube. Following an incubation period of 5 min, the $^1$H NMR spectra was obtained. Ethylene concentrations in solution were determined through integration and comparison with the internal standard.

**Dose-dependent reactivity of RuQ with ethylene gas**. For dose-dependent reactivity experiments, a 100 μM solution of **RuQ** in solvent (either 100% THF or 10% THF in H$_2$O) was prepared in a septum-sealed cuvette. Before injection, a measurement was recorded ($t = 0$ min). A specific volume of ethylene gas (1000, 500, 250, 100, 50 μl) was then quickly bubbled into solution. The cuvette was incubated for 1 min before another measurement was recorded ($t = 1$ min). Fluorescence measurements were made using a JASCO FP-6500 spectrofluorometer ($\lambda_{EX} = 420$ nm/$\lambda_{EM} = 463$ nm). All reactivity studies were performed in duplicate from distinct samples.

**Time-based reactivity of AEP with ethylene gas**. For time-based monitoring of **AEP** reactivity, water (1425 μl) was added to a septum-sealed cuvette and purged with a balloon of ethylene gas. A solution of **AEP** (75 μl from 2 mM stock solution in H$_2$O) was then added via syringe. The cuvette was then incubated at r.t. and monitored at set time intervals (10, 30, 60, 120, and 180 min). Fluorescence measurements were made using a JASCO FP-6500 spectrofluorometer ($\lambda_{EX} = 420$ nm/$\lambda_{EM} = 463$ nm). All reactivity studies were performed in triplicate from distinct samples.

**AEP-selectivity assay**. To investigate **AEP** reactivity with various terpenes, water (1275 μl) was first added to a cuvette, followed by a solution of **AEP** (75 μl from 2 mM stock solution in H$_2$O) and various terpenes (150 μl from 60 mM stock solution in THF). The cuvette was then incubated at r.t. and monitored at 60 min. Fluorescence measurements were made using a JASCO FP-6500 spectrofluorometer ($\lambda_{EX} = 420$ nm/$\lambda_{EM} = 463$ nm). All reactivity studies were performed in triplicate from distinct samples.

**Spatial imaging with kiwifruit**. Unripe and ripening kiwifruit were obtained from the local supermarket. Sections of kiwifruit were prepared using a kitchen knife at rough dimensions of about 2.0 cm × 4.5 cm, where the aim was to obtain slices that included the outer pericarp, locules, and the columella of the fruit. To monitor ethylene production, 170 μl of **AEP** (400 μM solution) was first applied to the center of a 10 cm Petri dish. Samples were then gently placed on top of the **AEP** solution to ensure even exposure. Samples were left to incubate at r.t., where imaging was carried at certain time intervals (1, 24 h) using a Keyence BZ-X710 All-in-one Fluorescence Microscope® equipped with a ET-EBFP2/Coumarin/Attenuated DAPI Filter Set Cat#49021 (Chroma Technology Corp., Vermont, USA). Brightfield images (color) were obtained at a 1/25 s exposure setting and fluorescent images were obtained at a 1/3.5 s exposure setting. Multiple images were obtained at ×4 magnification, where image stitching and analysis was performed using BZ-X Analyzer software (Keyence, Japan).

**Spatial imaging with Asian pears**. Asian pears were obtained from the local supermarket. Sections of flesh (hyphanthium tissue) were prepared using a kitchen slicer, at rough dimensions of about 4.8 cm × 4.8 cm. Samples were then positioned in the center of a 10 cm Petri dish, where cotton balls were placed on the sample surface and allowed to incubate for 5 h at r.t. Four of the five cotton balls are soaked with water, while the remaining cotton ball is soaked with either 10 mM ACC or 10 mM PZA. To monitor ethylene production, 300 μl of **AEP** (400 μM solution) was first applied to the center of another 10 cm Petri dish. Samples were then gently placed on top of the **AEP** solution to ensure even exposure. Samples were left to incubate at r.t., where imaging was carried out after 18 h using a Keyence BZ-X710 All-in-one Fluorescence Microscope® equipped with a ET-EBFP2/Coumarin/Attenuated DAPI Filter Set Cat#49021 (Chroma Technology Corp., Vermont, USA). Brightfield images (color) were obtained at a 1/25 s exposure setting and fluorescent images were obtained at a 1/3.5 s exposure setting. Multiple images were obtained at ×4 magnification, where image stitching and analysis was performed using BZ-X Analyzer software (Keyence, Japan).

**Time-dependent imaging with fruits**. Apples (Golden Delicious, Jazz, and Fuji cultivars), Asian pears, kiwifruit, and Muscat grapes, carrots, and red bell peppers were obtained from the local supermarket. Samples of tissue (small rectangles of ~10 mm × 8 mm) were prepared using a kitchen slicer and knife. Samples were then incubated in either water, ACC (1 or 10 mM), or PZA (1 or 10 mM) for 3 h at r.t. To monitor ethylene production, 10 μl of **AEP** (400 μM solution) was first applied to the center of individual wells of a chamber slide (ibidi μ-Slide eight-well chamber slides). Samples were then gently placed on top of the **AEP** solution to ensure even exposure. Samples were left to incubate at r.t., where imaging was carried in at certain time intervals (0, 1, 2, 4, 8, 20 h) using a Keyence BZ-X710 All-in-one Fluorescence Microscope® equipped with a ET-EBFP2/Coumarin/Attenuated DAPI Filter Set Cat#49021 (Chroma Technology Corp., Vermont, USA). Brightfield images (color) were obtained at a 1/80 s exposure setting and fluorescent images were obtained at a 1/6 s exposure setting. Multiple images were obtained at ×4 magnification, where image stitching and analysis was performed using BZ-X Analyzer software (Keyence, Japan).

**Imaging with *A. thaliana* plants**. *A. thaliana* plants were grown on soil at 23 °C and a light intensity of 85 μmol m$^{-2}$ s$^{-1}$. A 10 h light and 14 h dark photoperiod

was applied. To monitor ethylene production, leaves were first harvested from 4 to 6 weeks old plants. Using forceps, clear epidermal peels were then removed from leaves and placed into 96-well clear bottom plates. Following the addition of $H_2O$ (100 μl), the samples were then incubated for 12 h at r.t. to cancel wound stress. Depending on the conditions under study, various solutions were added as necessary: 1 mM of ACC (2 μl of 55 mM stock solution in $H_2O$); 4.8 μM of flg22 or elf18 (5 μl of 100 μM stock solution in $H_2O$); or $OD_{600} = 0.02$ solution of *Pseudomonas* bacteria (2 μl of $OD_{600} = 1.0$ stock solution). For studies involved with ACC and *Pseudomonas* bacteria, an incubation period of 12 h at r.t. was applied. For studies involved with PAMPs, an incubation period of 6 h at r.t. was instead applied. The solutions were then completely removed from the epidermal peel samples, followed by the addition of 100 μM **AEP** (50 μl stock solution). After 30 min, solutions were completely removed, washed with $H_2O$, and then imaged on a Keyence BZ-X710 All-in-one Fluorescence Microscope® equipped with a ET-EBFP2/Coumarin/Attenuated DAPI Filter Set Cat#49021 (Chroma Technology Corp., Vermont, USA). Images were obtained at ×20 and ×40 magnification, where brightfield images (monochrome) were obtained at a 1/400 s exposure setting and fluorescent images were obtained at a 1/30 s exposure setting. Analysis was performed using BZ-X Analyzer software (Keyence, Japan).

**Reporting summary**. Further information on research design is available in the Nature Research Reporting Summary linked to this article.

## Data availability

All data supporting the findings of this study are available with the article, and can also be obtained from the corresponding author upon reasonable request. The source data underlying Figs. 2b–g; 3b; 4e; 5b–e; 6a–b, d–e, g–h; 7e; 8f; 9f, and Supplementary Figs. 42a, b, d, e are provided as a Source Data file.

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

## Acknowledgements

Kind acknowledgement is given to GlyTech Inc. for supporting this research. CD spectral measurements were acquired by the Molecular Structure Characterization Unit, RIKEN Center for Sustainable Resource Science (CSRS). This work was financially supported by the MEXT/JSPS KAKENHI Grant Numbers JP16H03287, JP18K19154, JP15H05843 (to K.T.), JP16H06186 (to Y.K.), JP15H05959 (to K. S.), and JP18K14347 (to K.V.). This work was also supported by a RIKEN Incentive Research Project grant to I.N. and Y.K., as well as the Russian Government Program for Competitive Growth (granted to Kazan Federal University).

## Author contributions

Preparation of reagents was done by S.E., K.V., and I.N. Preparation of plant and bacterial strains were carried out by Y.K. Fruit-imaging studies were carried out by K.V., while plant-imaging studies were done by S.E. and T.W. Reactivity and spectroscopic studies were performed by K.V. and S.E. The manuscript was written by K.V. and K.T., and checked by S.Y. and K.S. This research was directed and supervised by K.T.

## Competing interests

The authors declare no competing interests.
