## [Peer Review File · Nature Communications]

Reviewers' comments:

Reviewer #1 (Remarks to the Author):

Summary

The manuscript entitled "Artificial Metalloenzyme Biosensor of Ethylene Gas in Fruits and Leaves of Plants" is a well-written description of an impressive body of work relating to the development of an ethylene probe for in vivo measurements of ethylene in plants. The authors have constructed their probe following on from their past work using Human serum albumin as a proteinaceous scaffold to solubilize and protect bound catalytic metals. In this case, albumin harbors a fluorescent DEAC-Ru conjugated to and quenched by a DABCYL moiety. This DEAC-Ru-DABCYL complex within albumin is termed AEP and becomes fluorescent in the presence of ethylene gas, which has high reactivity for olefin metathesis and thereby displaces the DABCYL quencher from fluorescent DEAC-Ru. This process is irreversible and therefore the probe can only be used to examine steady state or increases in ethylene, though this is still potentially very useful. The authors perform a series of proof-of-principle experiments in apple fruits and Arabidopsis leaves with increased or decreased ethylene production, thus convincingly demonstrating the potential for AEP to serve as a new tool in the plant science and agronomy fields for in vivo measurements of ethylene. This is particularly valuable as ethylene has no comparable in vivo sensors available and, as a gas, poses particular challenges for in vivo quantification.

Major comments

1. The authors claim that they have created a tool that can measure ethylene in planta with spatial resolution not previously possible. However, I am not entirely convinced that they have shown this to be true. The only spatially resolved data they have presented is that the interior of mashed apple fruit sections has more AEP fluorescence than unmashed sections. The authors conclude that this results from increased ethylene production, but could this simply be better accessibility for the AEP probe to the ethylene in the tissues? I suggest that the authors do more to address the questions of 1. How quickly the AEP probe enters plant tissues and 2. How far into and into which type of plant tissues the probe can travel?
2. Alternatively, and more interestingly, the authors could set up an experiment where ethylene should be more localised and use their tool to detect ethylene with high spatial resolution. For example, could flg22 be delivered to part of a leaf and then the spatial distribution of ethylene production be examined?
3. The authors similarly claim that they have a tool that can measure ethylene production in higher temporal resolution, but the only experiment they do here could have been done more effectively using GC-MS methods. Namely, they measured ethylene production at 1hr and 9hr endpoints. I suggest the authors do more to demonstrate the temporal utility of their probe. For example, can they detect ethylene increases as they happen (i.e. with AEP pre-loaded onto a sample)? Also, for how long does the AEP probe remain viable when loaded onto plant tissues?
4. The sensitivity of the probe remains unclear – why not have a dose response curve at least in vitro?
5. Could toxicity be an issue here? Perhaps unlikely, but would be nice to know if the probe itself is broadly harmful to plant physiology and development.
6. The authors claim they have built an ethylene probe 'from the ground up' and therefore this work presents a more general approach for "biosensors composed of abiotic metal complexes, where the bioorthogonality of certain transition metals can help to target metabolites found in difficult-to-access chemical spaces". It seems to me the key here was being able to quench DEAC-Ru with an ethylene sensitive DABCYL olefin. Will it truly be so straightforward to make DEAC-Ru fluorescence sensitive to other compounds? A steroid hormone, for example? This is not to say that the approach has not been successful, just that I'm not clear on the claims of generality.

Minor comments:

1. The paragraph describing eto1 function is awkward and should be revised.

2. PPR receptor --- PRR
3. Generally the writing is very clear, only a few typos that need further copyediting

Reviewer #2 (Remarks to the Author):

The work of Eda et al is an interesting paper that describes the development of a fluorescence based ethylene biosensor that is comprised of an artificial metalloenzyme. They have developed an enzymatic sensor that harbors a fluorescent DEAC-Ru complex that is quenched by DABCYL in the hydrophobic pocket. This quenching is lost upon reactivity with ethylene, restoring the fluorescence signal. This artificial ethylene sensor is called AER. The functionality of this sensor was tested using the ethylene releasing agent ethephon and several bioassays (damaged apple tissue, Arabidopsis leaf peels of different mutants and leaf peels infected with effectors and pathogens).

I believe the development of such a functional ethylene sensor is long sought by the community, and could be a major breakthrough in the plant science field. This novel ethylene sensor could be extremely valuable to the ethylene and postharvest community for research or commercial applications.

Yet, the work of Seda et al fails to provide sufficient evidence of the specificity of the sensor. Critical for any sensor development study is to test specificity, sensitivity and affinity. I suggest the authors add several (in vitro) assays to validate the specificity for ethylene gas (by using ethylene gas and not ethephon), the sensitivity (dose response curves, level of detection) and binding affinity (Kon and Koff kinetics) for the AER sensor. This characterization would provide essential physicochemical properties of the sensor, and make the biological case-studies more trustworthy. Furthermore, I wonder if the sensor can be reverted. So, when ethylene gas is not present anymore, is then the DABCYL quenching restored and thus the fluorescence signal lost again, or not? This is critical to know in order to use the sensor in time-course or dynamic experiments.

They also mention several times that they have developed the first ethylene sensor that allows users to visualize ethylene in a spatiotemporal way. They forgot about the extremely useful and widely used EBS-GUS or EBS-GFP reporter lines. They should incorporate and/or discuss this EBS reporter line in comparison to their EAR sensor. Furthermore, the spatiotemporal aspect of the AER sensor could be proven much better if tissue specificity in ethylene production could have been showed (thus look at multiple tissues). This would have strengthened the statement that their AER sensor is a first sensor that has spatial resolution, as the EBS-lines have.

Below are more details on some minor and major concerns/questions.

It would have been easy to have line numbering to aid the reviewing process.

P2: the authors mention that trace amounts of ethylene is produced in unripen and pathogen-free fruits and vegetables. But this statement is shortsighted. Ethylene levels can dramatically increase in unripe and pathogen-free tissue of all kinds of plant organs (not only fruits and vegetables) due to environmental or developmental stimuli. So I advise, to modify this statement of trace amounts (in the field this is called "basal ethylene from System I production") to specific processes and to plants in general (not only fruits and vegetables).

P2: the authors say that numerous studies have shown that the processes of abscission and ripening can be accelerated by ethylene, yet they only provide one reference (dealing only with ripening). Perhaps more leading papers can be cited.

P2: The authors mention three techniques to measure ethylene (gas chromatography, electrochemical and laser-based techniques) and mention that spatiotemporal resolution is wanted. I agree, but the authors forget to mention the existing bio-sensor: EBS::GUS or EBS::GFP transgenic lines. The Stepanova group has made these lines a long time ago, and distributed them worldwide for many years. These EBS reporter lines have become the most used ethylene visualization technique to date, have spatio-temporal resolution, and thus deserve a proper

mentioning in this paper.

Figure 1 shows the chosen approach, but does not address the potential of the spatiotemporal applications of the new biosensor. This could be incorporated in Figure 1.

Figure 2D. The authors used ethephon to release ethylene to test their AER sensor in vitro. I wonder what the pH was of the solution, because the release of ethylene from ethephon is known to be pH sensitive? This is not mentioned in the text or the experimental section. The drawback of using ethephon is that one is never sure of its specificity.

The authors should provide a positive control in which they use pure ethylene gas of a known concentration. Subsequently, other olefin gasses (alkenes, alkanes...) should be tested, to verify sensor specificity. Plants are known to produce a wide array of simple and complex organic gasses, which might interfere with the sensor. I wonder what the AER reporter would do if present in a headspace filled with ethylene or other plant produced gasses? Non-specific binding of other gasses/compounds to the AER sensor cannot be ruled out, unless specificity is demonstrated first. Furthermore, the authors should make a dose response curve of their AEP sensor using different concentrations of ethylene gas. This will allow us to know what is the level of detection (LoD?). A time-course using ethylene gas is also missing, and could tell us how reactive it is (K_m ?). The dissociation constant is also not investigated, and could tell us how long the fluorescence signal remains in the presence of ethylene (K_{off} ?).

I also wonder if it is possible that the DABCYL quencher molecule does not function anymore once ethylene has bound, and that the fluorescence signal is not lost over time when ethylene is not present anymore. In this case the sensor only gives an imprint of a certain ethylene production and is for sure not able to visualize ethylene in a time courses. This reversibility aspect of the AER sensor is not discussed in the manuscript. All together, I believe these are some mayor shortcomings, which could be incorporated by doing some simple additional experiments (in my opinion).

The authors present their AER sensor as a first spatio-temporal sensor for detecting ethylene production in plants. Although they provide examples with real biological samples (apple tissue and Arabidopsis leaf peels), they do not really demonstrate the spatial nor temporal potential of the sensor. They could maybe use an example to visualize tissue specific ethylene production (e.g. in the Arabidopsis primary root or in the developing tomato fruit tissues), which is also a temporal fingerprint for development and thus ethylene production. This approach (or similar), if possible, would make their sensor even more useful to the ethylene community, with respect to spatio-temporal resolution of ethylene detection.

P6: the authors mention that ACS is the rate limiting step in ethylene biosynthesis, yet they fail to cite a reference that states this. Recent insights in ethylene biosynthesis also highlight that this is not always the case, and that besides ACS, also ACO can be rate limiting, especially during post-climacteric ripening of fruit such as apple, a biological example the authors use in their study.

Figure 3: How do the authors explain the fact that the top side the control apple slices do not show any strong fluorescence signal compared to the sides? These were cut pieces of apple, so all sides were wounded, also the top side. If the sensor would detect wound ethylene, one would expect all sides of the slices to show a fluorescence signal. Furthermore, the normal photograph of the mashed apple tissue shows browning, which indicates oxidation has occurred that that mashing was intensive or that the sample was analyzed long after mashing. Other secondary metabolites could also have interacted with the AER sensor which were released during mashing. I believe wounding is not the best strategy to monitor ethylene production. There are many different examples of non-invasive plant processes that produce enormous amount of ethylene in which the AER sensor could have been tested, eliminating the need for destructive tissue preparations (wounding).

In case the sensor would detect ethylene, the wounding assay has another negative aspect. First of all it would be better if the authors can show that their wounding set-up results in tissue that is indeed producing more ethylene (by standard gas chromatography). Furthermore, it has been shown that wounding immediately released ethylene gas from the damaged tissue. This is the ethylene present in cells and their membranes, which is released. This is not produced by the cell itself. It takes about 10-20 min to get rid of this artifact. Then you have a time window of about 10 min when nothing happens, and than you gradually get more ethylene due to the wound response

(which is genetically programmed). The author should clearly mention which process of wound ethylene they are monitoring.

An alternative and better approach would have been feeding ACC to apple tissue, as they have done for their Arabidopsis assays. This boosts endogenous ethylene production without wounding the tissue. This would allow them to validate the biological usefulness of the AER sensor.

P7: The abbreviation of S-adenosyl-L-methionine (SAM; and not AdoMet) synthetase is SAMS and not MAT. Please use proper abbreviations.

P8: Is there any reason why the authors have used the Arabidopsis leaf peels? Does their AER sensor not work in leaves? I believe it is better to mention why they chose for a certain set-up. The *eto1* mutation does not negatively regulate ACS activity. It positively regulates ACS activity by not degrading Type 2 ACS. The *eto* mutation is in the F-box that recognizes the C-terminal ACS end, preventing it from degrading.

The authors used the *eto1/ckrc1* mutant. They explained the *eto1* background, but not the *ckrc1-1* background. This information should be provided. This mutation is in the auxin biosynthesis pathway, and it would have been better to solely use the *eto1* mutant or the *eto1/eol1/eol2* mutant, in which 3 ETO's (EOLS) that recognize ACS are non-functional.

Reviewer #3 (Remarks to the Author):

This is a very interesting and well-presented manuscript, in which the authors describe the development of an artificial metalloenzyme for the detection of ethene in apples and Arabidopsis thaliana leaves. The results are novel and fully supported by the experimental evidence given. Since artificial metalloenzymes and their application as biosensors are of much current interest, the manuscript merits publication in Nature Communications. There are, however, a few points that should be addressed:

1. In the title and throughout the text, please consider use of the IUPAC name ethene (instead of ethylene)
2. The ArM has only be tested in apples and Arabidopsis thaliana leaves, hence mentioning 'fruits and plant leaves' in the title is not justified; the title and abstract should be modified to accurately reflect scope of the study.
3. Page 2, last paragraph: copper(I) – remove space
4. Page 6: ... expected to proceed selectively.... It is correct that the olefin metathesis reactivity of ethene is high; however, the selectivity over other alkenes should be confirmed experimentally to ascertain that the presence of other unsaturated biomolecules, e.g. terpenes in plants, is not leading to false positives.
5. What is the limit of detection and the dynamic range in which the biosensor can be used?
6. Experimental section, page 15: Please state how the AEP was characterised (e.g. via mass spectrometry, absorption spectroscopy, circular dichroism etc.) and list the characterisation data obtained.
7. Experimental section, page 15: Fluorescence measurement: H₂O (use subscript) and $\lambda_{EX}=420\text{nm}/\lambda_{EM}=463\text{nm}$ (use subscript for EX and EM; add a space between numbers and the units)

1 COMMENT: The authors claim that they have created a tool that can measure ethylene in plants with spatial resolution not previously possible. However, I am not entirely convinced that they have shown this to be true. The only spatially resolved data they have presented is that the interior of mashed apple fruit sections has more AEP fluorescence than unmashed sections. The author's conclude that this results from increased ethylene production, but could this simply be better accessibility for the AEP probe to the ethylene in the tissues? I suggest that the author's do more to address the questions of 1. How quickly the AEP probe enters plant tissues and 2. How far into and into which type of plant tissues the probe can travel?

Alternatively, and more interestingly, the authors could set up an experiment where ethylene should be more localised and use their tool to detect ethylene with high spatial resolution. For example, could flg22 be delivered to part of a leaf and then the spatial distribution of ethylene production be examined?

RESPONSE: As this reviewer (and others) have pointed out, more experimental proof is needed to demonstrate the spatial utility of our probe. As such, a significant amount of effort was used to carry out two different types of spatial imaging. In one type of experiment, we focused on imaging endogenous levels of ethylene between ripening and unripe kiwifruit sections (Figure 3; Page 7-8, Line 181-197). From this data, we can see a clear increase in fluorescence in the outer pericarp of ripening kiwifruit compared to unripe kiwifruit.

In another type of experiment, we focused on spatial imaging of ethylene produced by sections of fruit tissue stimulated by an exogenous source of either ACC or PZA (following the advice of another reviewer comment). From this data, a clear increase or decrease in fluorescence can be seen in the vicinity where the stimulant was applied (Figure 4; Page 8-9, Line 206-222).

To answer the reviewer comment on probe penetrability, we have added a "Limitations" section that addresses this (Page 14, Line 345-365). In summary, the exterior of our probe is essentially an albumin protein (~66 kDa protein). Since proteins of this size do not passively cross the cell wall, it can be assumed that the AEP probe exists extracellularly, rather than intracellularly.

2 COMMENT: The authors similarly claim that they have a tool that can measure ethylene production in higher temporal resolution, but the only experiment they do here could have been done more effectively using GC-MS methods. Namely, they measured ethylene production at 1hr and 9hr endpoints. I suggest the authors do more to demonstrate the temporal utility of their probe. For example, can they detect ethylene increases as they happen (i.e. with AEP pre-loaded onto a sample)? Also, for how long does the AEP probe remain viable when loaded onto plant tissues?

RESPONSE: We would like to thank the reviewer for this constructive comment. To address the issue of temporal imaging, a series of experiments were added to this manuscript (Figures 5-6; Page 9-10, Line 224-246). In summary, several different fruit samples were placed onto slides with pre-loaded AEP and imaged at the same spot over several timepoints. The addition of these experiments should better demonstrate the temporal usage of our probe.

3 COMMENT: The sensitivity of the probe remains unclear – why not have a dose response curve at least in vitro?

RESPONSE: In general, there are severe limitations of working with ethylene gas, largely due to its poor water-solubility. An example of this is best highlighted by various dose response curves obtained for RuQ (Figures 2e; Page 6, Line 144-157). In pure organic solvent, good reactivity can be seen because injected ethylene was largely found to be soluble, which is supported by NMR solubility studies (Supplementary Figure 1). However, under aqueous conditions, RuQ reactivity appears to be dramatically lowered. This is in fact due to the poor water solubility of ethylene, where NMR solubility studies found no detectable levels of ethylene dissolved in solution. Given these challenges, it is difficult to devise an experimental approach to construct dose response curves for AEP using known concentrations of dissolved ethylene gas under aqueous conditions.

In order to design an experimental setup to show direct reaction of ethylene with the AEP probe

under aqueous conditions, the following assumptions were made. According to Henry's gas law, the amount of dissolved gas in a liquid is proportional to its partial pressure above the liquid. As such, one setup could see a closed system (with the cuvette) containing water that is purged and fitted with a balloon containing ethylene gas. In this manner, it can be assumed that the only gas present is ethylene. Since the pressure of the attached balloon does not significantly deviate from 1 atm, it can be assumed that the total pressure of ethylene gas present is equivalent to its partial pressure. Using a Henry's law constant of 5.98×10^{-3} M/atm, the calculated concentration of dissolved ethylene in water would thus be approximately 6 mM, or 167 ppm. To initiate time-dependent reactivity studies, AEP was then injected into the ethylene-filled cuvette and monitored over time (Figures 2f; Page 6-7, Line 158-169). As shown, a significant time-dependent increase in fluorescence was observed compared to controls.

4 COMMENT: Could toxicity be an issue here? Perhaps unlikely, but would be nice to know if the probe itself is broadly harmful to plant physiology and development.

RESPONSE: Given that the exterior of our AEP probe is an albumin protein, we also do not believe that its addition to plants will be harmful.

In terms of experimental proof, some insight may be offered by alb-Ru (structurally depicted in Figure 2). This similar protein complex was tested for cytotoxicity against eukaryotic cells in a previous publication by our group. From those results, alb-Ru was not found to be cytotoxic. As such, it is very likely that the AEP probe will also be non-cytotoxic to plants.

5 COMMENT: The authors claim they have built an ethylene probe 'from the ground up' and therefore this work presents a more general approach for "biosensors composed of abiotic metal complexes, where the bioorthogonality of certain transition metals can help to target metabolites found in difficult-to-access chemical spaces". It seems to me the key here was being able to quench DEAC-Ru with an ethylene sensitive DABCYL olefin. Will it truly be so straightforward to make DEAC-Ru fluorescence sensitive to other compounds? A steroid hormone, for example? This is not to say that the approach has not been successful, just that I'm not clear on the claims of generality.

RESPONSE: We apologize if we are misinterpreting the reviewer's question, but in general, olefin metathesis would only be applicable to ethylene detection. Unfortunately, we do not think DEAC-Ru fluorescence can be fine-tuned to become sensitive to other compounds (i.e. steroid hormone). We think the source of the confusion is that we stated ArM-based biosensors can be potentially applied for other biological metabolites. However, in this case, we would need to change out the metal catalyst (i.e. palladium, gold, iridium, etc) to carry out other reactions applicable to the target metabolite.

In terms of selectivity, this reviewer may be interested to know that we have added a selectivity assay to our manuscript to demonstrate ethylene selectivity (Figure 2g; Page 7, Line 170-179). In this test, various terpenes were incubated with our AEP probe. Overall, we saw greater reactivity when using ethylene gas.

6 COMMENT: The paragraph describing eto1 function is awkward and should be revised.

RESPONSE: We apologize over our previous confusion concerning the eto1 plant mutant. As requested, we have updated the paragraph of interest (Page 10, Lines 256-260).

7 COMMENT: PPR receptor --- PRR

RESPONSE: We would like to thank the reviewer for this correction. As advised, we have made the indicated changes (Page 11, Line 290; Page 12, Line 294)

----- *Reviewer 2* -----

8 COMMENT: the authors mention that trace amounts of ethylene is produced in unripen and pathogen-free fruits and vegetables. But this statement is shortsighted. Ethylene levels can dramatically increase in unripe and pathogen-free tissue of all kinds of plant organs (not only fruits and vegetables) due to environmental or developmental stimuli. So I advise, to modify this statement of trace amounts (in the field this is called “basal ethylene from System I production”) to specific processes and to plants in general (not only fruits and vegetables).

RESPONSE: We would like to thank the reviewer for bringing our attention to this part of the text. Indeed, our wording was poor and lacked the necessary information/referencing. As requested, we have updated this text with a more accurate explanation (Page 2, Lines 43-47).

9 COMMENT: the authors say that numerous studies have shown that the processes of abscission and ripening can be accelerated by ethylene, yet they only provide on reference (dealing only with ripening). Perhaps more leading papers can be cited.

RESPONSE: We would like to apologize for our poor effort in terms of referencing at this part of the text. As advised, we have attempted to include more leading papers related to fruit abscission/ripening from exogenous ethylene sources (Page 2, Lines 47-48).

10 COMMENT: The authors mention three techniques to measure ethylene (gas chromatography, electrochemical and laser-based techniques) and mention that spatiotemporal resolution is wanted. I agree, but the authors forget to mention the existing bio-sensor: EBS::GUS or EBS::GFP transgenic lines. The Stepanova group has made these lines a long time ago, and distributed them worldwide for many years. These EBS reporter lines have become the most used ethylene visualization technique to date, have spatio-temporal resolution, and thus deserve a proper mentioning in this paper.

RESPONSE: As researchers outside of the plant science field, we find these comments extremely helpful to our understanding. We would like to sincerely thank the reviewer for bringing our attention to this work, which we were previously unaware about. As advised, we have attempted to give the EBS:GUS plant lines a proper mention in our manuscript (Page 2, Line 55-56). If there are any other critical works we have missed, please do not hesitate to let us know.

11 COMMENT: Figure 1 shows the chosen approach, but does not address the potential of the spatiotemporal applications of the new biosensor. This could be incorporated in Figure 1.

RESPONSE: As kindly pointed out by the reviewer, we have updated Figure 1 to show that our aim is to develop an ethylene probe with potential for spatiotemporal detection.

12 COMMENT: Figure 2D. The authors used ethephon to release ethylene to test their AER sensor in vitro. I wonder what the pH was of the solution, because the release of ethylene from ethephon is known to be pH sensitive? This is not mentioned in the text or the experimental section. The drawback of using ethephon is that one is never sure of its specificity.

RESPONSE: Due to the request (from all reviewers) that AEP reactivity be shown directly using ethylene gas, we felt it would be best to completely remove the data regarding ethephon from this manuscript. However, to answer the reviewer's question, we performed the ethephon study using PBS Buffer pH 7.4. while also heating on a 37°C water bath. We apologize for this carelessly omission of the experimental setup.

13 COMMENT: The authors should provide a positive control in which they use pure ethylene gas of a known concentration. Subsequently, other olefin gasses (alkenes, alkanes...) should be tested, to verify sensor specificity. Plants are known to produce a wide array of simple and complex organic gasses, which might interfere with the sensor. I wonder what the AER reporter would do if present in a headspace filled with ethylene or other plant produced gasses? Non-specific binding of other gasses/compounds to the AER sensor cannot be ruled out, unless specificity is demonstrated first.

Furthermore, the authors should make a dose response curve of their AEP sensor using different concentrations of ethylene gas. This will allow us to know what is the level of detection (LoD?). A time-course using ethylene gas is also missing, and could tell us how reactive it is (Km?). The dissociation constant is also not investigated, and could tell us how long the fluorescence signal remains in the presence of ethylene (Koff?).

RESPONSE: In attempts to carry out the requested experiments, we ultimately came to realize the experimental limitations of working with ethylene gas, which is known to display poor water-solubility. In our initial trials (Figures 2e; Page 6, Line 144-157), we obtained a dose response curve for RuQ in pure organic solvent. In this case, ethylene injected into solution was soluble, as highlighted in NMR solubility studies (Supplementary Figure 1). However, when we performed an equivalent dose-response study of RuQ under aqueous conditions, a dramatic decrease in reactivity was observed. Looking closely at NMR solubility studies, we found that there were no detectable levels of ethylene dissolved in solution. Given these conditions, experiments using injections of known volumes of ethylene gas thus become infeasible. Likely, a vast majority of the injected ethylene remains in the headspace above the liquid. We also did not want to increase the injection volume beyond 1 ml of gas, as built-up pressure in the cuvette (made of glass) presents itself as a safety hazard.

In order to design an experimental setup that can ensure an adequate level of ethylene is present under aqueous conditions to test AEP reactivity, the following assumptions were made. According to Henry's gas law, the amount of dissolved gas in a liquid is proportional to its partial pressure above the liquid. As such, one setup could see a closed system (with the cuvette) containing water that is purged and fitted with a balloon containing ethylene gas. In this manner, it can be assumed that the only gas present is ethylene. Since the pressure of the attached balloon does not significantly deviate from 1 atm, it can be assumed that the total pressure of ethylene gas present is equivalent to its partial pressure. Using a Henry's law constant of 5.98×10^{-3} M/atm, the calculated concentration of dissolved ethylene in water would thus be approximately 6 mM, or 167 ppm. To initiate time-dependent reactivity studies, AEP was then injected into the ethylene-filled cuvette and monitored over time (Figures 2f; Page 6-7, Line 158-169). As shown, a significant time-dependent increase in fluorescence was observed compared to controls.

It should be noted that one fault of this experimental setup is that the concentration of dissolved ethylene cannot be controlled, as we are essentially saturating the aqueous solution with ethylene. We surmise that without equipment to properly monitor and control the partial pressure of ethylene above the liquid (thus giving us a concentration in liquid), it is experimentally difficult to obtain a dose response curve using AEP under aqueous conditions.

As this reviewer (and others) have pointed out, more experimental proof is needed to demonstrate the selectivity of our probe. In response, we have added a selectivity assay to our manuscript (Figure 2g; Page 7, Line 170-179). In this test, various terpenes were incubated with our AEP probe.

14 COMMENT: I also wonder if it is possible that the DABCYL quencher molecule does not function anymore once ethylene has bound, and that the fluorescence signal is not lost over time when ethylene is not present anymore. In this case the sensor only gives an imprint of a certain ethylene production and is for sure not able to visualize ethylene in a time courses. This reversibility aspect of the AER sensor is not discussed in the manuscript.

RESPONSE: We would like to thank the reviewer for this comment, as it is an excellent point to be brought up. In short, once the DABCYL quencher leaves the albumin binding pocket, it should be much more difficult to re-enter the albumin binding pocket under a complex biological environment. In addition, the DABCYL quencher would need to continue to compete against the more reactive ethylene gas. A small discussion about the reversibility aspect has now been added to the manuscript (Page 6, Line 141-143).

15 COMMENT: The authors present their AER sensor as a first spatio-temporal sensor for detecting ethylene production in plants. Although they provide examples with real biological samples (apple tissue and Arabidopsis leaf peels), they do not really demonstrate the spatial nor temporal potential of the sensor. They could maybe use an example to visualize tissue specific ethylene production (e.g. in the Arabidopsis primary root or in the developing tomato fruit tissues), which is also a temporal fingerprint for development and thus ethylene production. This approach (or similar), if possible, would make their sensor even more useful to the ethylene community, with respect to spatio-temporal resolution of ethylene detection.

RESPONSE: Considering all reviewers have pointed out that more experimental proof for spatial and temporal imaging was necessary, we took this opportunity to pursue various experiments based on the advice we received.

Spatial Imaging: Since cross sections were difficult to obtain for tomato fruit (using our equipment), we settled on using kiwifruit, which is also known to experience an increase in ethylene production in its outer pericarp during ripening (Figure 3; Page 7-8, Line 181-197). From this data, we can clearly see an increase in outer pericarp fluorescence in ripening kiwifruit compared to unripe kiwifruit.

Temporal Imaging: To address the issue of temporal imaging, a series of experiments were added to this manuscript (Figures 5-6; Page 9-10, Line 224-246). In summary, several different fruit samples were placed onto slides with pre-loaded AEP and imaged at the same spot over several timepoints. The addition of these experiments should better demonstrate the temporal usage of our probe.

16 COMMENT: the authors mention that ACS is the rate limiting step in ethylene biosynthesis, yet they fail to cite a reference that states this. Recent insights in ethylene biosynthesis also highlight that this is not always the case, and that besides ACS, also ACO can be rate limiting, especially during post-climacteric ripening of fruit such as apple, a biological example the authors use in their study.

RESPONSE: Since this is relevant information that we were not aware about, we would like to thank the reviewer for kindly bringing our attention to this. As requested, we have updated this text with a more accurate explanation and provided references which we hope will satisfy the reviewer (Page 8, Lines 200-202).

17 COMMENT: How do the authors explain the fact that the top side the control apple slices do not show any strong fluorescence signal compared to the sides? These were cut pieces of apple, so all sides were wounded, also the top side. If the sensor would detect wound ethylene, one would expect all sides of the slices to show a fluorescence signal. Furthermore, the normal photograph of the mashed apple tissue shows browning, which indicates oxidation has occurred that that mashing was intensive or that the sample was analyzed long after mashing. I believe wounding is not the best strategy to monitor ethylene production. There are many different examples of non-invasive plant processes that produce enormous amount of ethylene in which the AER sensor could have been tested, eliminating the need for destructive tissue preparations (wounding). In case the sensor would detect ethylene, the wounding assay has another negative aspect. First of all it would be better if the authors can show that their wounding set-up results in tissue that is indeed producing more ethylene (by standard gas chromatography). Furthermore, it has been shown that wounding immediately released ethylene gas from the damaged tissue. This is the ethylene present in cells and their membranes, which is released. This is not produced by the cell itself. It takes about 10-20 min to get rid of this artifact. Then you have a time window of about 10 min when nothing happens, and than you gradually get more ethylene due to the wound response. The author should clearly mention which process of wound ethylene they are monitoring.

RESPONSE: We would like to sincerely thank the reviewer for their insightful comments regarding the negatives associated with plant wounding experiments. Considering the need to obtain better spatial imaging data to convince all reviewers, we took this chance to pursue other experimental setups that did not rely on plant wounding. As a result of our newly acquired data, we feel it is now unnecessary to include our previous plant wounding data into this manuscript (especially considering the flaws pointed out by the reviewer). Again, we would like to thank the reviewer for the kind direction.

18 COMMENT: An alternative and better approach would have been feeding ACC to apple tissue, as they have done for their Arabidopsis assays. This boosts endogenous ethylene production without wounding the tissue. This would allow them to validate the biological usefulness of the AER sensor.

RESPONSE: As kindly suggested by this reviewer, we took the opportunity to follow this advice and perform spatial imaging on sections of fruit tissues that have been stimulated with either ACC or PZA. From this data, a clear increase or decrease in fluorescence can be seen in the vicinity where the stimulant was first applied (Figure 4; Page 8-9, Line 206-222).

19 COMMENT: The abbreviation of S-adenosyl-L-methionine (SAM; and not AdoMet) synthetase is SAMS and not MAT. Please use proper abbreviations.

RESPONSE: Our choice of nomenclature came from BRENDA, a highly respected enzyme database that uses information classified by the IUBMB. The enzyme known as EC 2.5.1.6 has an official name of "methionine adenosyltransferase" with MAT being an accepted abbreviated form. Having said this, we understand this request by the reviewer as it appears to be the standard in the plant science community. As advised, we have made this naming change (Figure 4; Page 8 Line 199).

20 COMMENT: Is there any reason why the authors have used the Arabidopsis leaf peels? Does their AER sensor not work in leaves? I believe it is better to mention why they chose for a certain set-up.

RESPONSE: The main issue with our imaging equipment is that we only have access to a typical inverted fluorescence microscope. Since plant leaves are curved by nature and do not easily conform to a single plane, it was difficult to find the right plane of view for fluorescence imaging (some spots would be in focus while other spots would be out of focus). To answer the reviewer's comment, our choice of analysing epidermal leaf peels stems mainly from its translucency and ability to conform to the flat surface of microscope imaging slides.

21 COMMENT: The eto1 mutation does not negatively regulate ACS activity. It positively regulates ACS activity by not degrading Type 2 ACS. The eto mutation is in the F-box that recognizes the C-terminal ACS end, preventing it from degrading.

The authors used the eto1/ckrc1 mutant. They explained the eto1 background, but not the ckrc1-1 background. This information should be provided. This mutation is in the auxin biosynthesis pathway, and it would have been better to solely use the eto1 mutant or the eto1/eol1/eol2 mutant, in which 3 ETO's (EOLS) that recognize ACS are non-functional.

RESPONSE: We would like to thank the reviewer for bringing our attention to this significant mistake on our part concerning the explanation of the eto1 plant mutant. As requested, we have updated the paragraph accordingly (Page 10, Lines 256-260).

In addition, the reviewer's comment also compelled us to acquire the single eto1-1 plant mutant. Using this sample, imaging experiments under the same conditions were shown to give approximately the same data. As such, images and data in Figure 7 were updated accordingly.

----- *Reviewer 3* -----

22 COMMENT: In the title and throughout the text, please consider use of the IUPAC name ethene (instead of ethylene)

RESPONSE: As chemists, we certainly agree this to be a reasonable request by the reviewer. Having said that, we have been informed from our plant biology collaborators that ethylene is the more common terminology used in their field. Given that this work is geared towards and has more implications to the plant biology community, we collectively felt it is better to continue using the term “ethylene”.

We have added the IUPAC term “ethene” at its first mention in the introduction (Page 2, Line 41). Although we could not completely address this concern, we would like to thank this reviewer for the suggestion.

23 COMMENT: The ArM has only be tested in apples and Arabidopsis thaliana leaves, hence mentioning ‘fruits and plant leaves’ in the title is not justified; the title and abstract should be modified to accurately reflect scope of the study.

RESPONSE: The reviewer is absolutely correct to point out the overstatement of our results. During the revision of this manuscript, we were able to test several other fruits for this study. As a result, we have changed our title to “fruits and A. thaliana leaves”.

24 COMMENT: Page 2, last paragraph: copper(I) – remove space

RESPONSE: We would like to thank the reviewer for this comment. As advised, we have made the indicated change (Page 3, Line 63).

25 COMMENT: Page 6: ... expected to proceed selectively.... It is correct that the olefin metathesis reactivity of ethene is high; however, the selectivity over other alkenes should be confirmed experimentally to ascertain that the presence of other unsaturated biomolecules, e.g. terpenes in plants, is not leading to false positives.

RESPONSE: As this reviewer (and others) have pointed out, more experimental proof is needed to demonstrate the selectivity of our probe. In response, we have added a selectivity assay to our manuscript (Figure 2g; Page 7, Line 170-179). In this test, various terpenes were incubated with our AEP probe. Although there is some reactivity present for terminal alkene-containing molecules (i.e. limonene, myrcene), we generally saw greater overall reactivity using ethylene gas. We surmise the main reason for selectivity comes from the properties of the albumin binding pocket where our catalyst is anchored into. We thank the reviewer for bringing our attention to this crucial experiment.

26 COMMENT: What is the limit of detection and the dynamic range in which the biosensor can be used?

RESPONSE: In attempts to carry out the requested experiments, we ultimately came to realize the experimental limitations of working with ethylene gas, which is known to display poor water-solubility. In our initial trials (Figures 2e; Page 6, Line 144-157), we obtained a dose response curve for RuQ in pure organic solvent. In this case, ethylene injected into solution was soluble, as highlighted in NMR solubility studies (Supplementary Figure 1). However, when we performed an equivalent dose-response study of RuQ under aqueous conditions, a dramatic decrease in reactivity was observed. Looking closely at NMR solubility studies, we found that there were no detectable levels of ethylene dissolved in solution. Given these conditions, experiments using injections of known volumes of ethylene gas thus become infeasible. Likely, a vast majority of the injected ethylene remains in the headspace above the liquid. We also did not want to increase the injection volume beyond 1 ml of gas, as built-up pressure in the cuvette (made of glass) presents itself as a safety hazard.

In order to design an experimental setup that can ensure an adequate level of ethylene is present under aqueous conditions to test AEP reactivity, the following assumptions were made. According to

Henry's gas law, the amount of dissolved gas in a liquid is proportional to its partial pressure above the liquid. As such, one setup could see a closed system (with the cuvette) containing water that is purged and fitted with a balloon containing ethylene gas. In this manner, it can be assumed that the only gas present is ethylene. Since the pressure of the attached balloon does not significantly deviate from 1 atm, it can be assumed that the total pressure of ethylene gas present is equivalent to its partial pressure. Using a Henry's law constant of 5.98×10^{-3} M/atm, the calculated concentration of dissolved ethylene in water would thus be approximately 6 mM, or 167 ppm. To initiate time-dependent reactivity studies, AEP was then injected into the ethylene-filled cuvette and monitored over time (Figures 2f; Page 6-7, Line 158-169). As shown, a significant time-dependent increase in fluorescence was observed compared to controls.

It should be noted that one fault of this experimental setup is that the concentration of dissolved ethylene cannot be controlled, as we are essentially saturating the aqueous solution with ethylene. We surmise that without equipment to properly monitor and control the partial pressure of ethylene above the liquid (thus giving us a concentration in liquid), it is experimentally difficult to obtain a dose response curve using AEP under aqueous conditions.

27 COMMENT: Experimental section, page 15: Please state how the AEP was characterised (e.g. via mass spectrometry, absorption spectroscopy, circular dichroism etc.) and list the characterisation data obtained.

RESPONSE: We apologize for the missing information. As requested, we have stated in the AEP preparation section how our probe was characterized (Page 18, Lines 435). In terms of characterization data, this is shown in Figures 2b-d.

28 COMMENT: Experimental section, page 15: Fluorescence measurement: H₂O (use subscript) and $\lambda_{EX}=420\text{nm}/\lambda_{EM}=463\text{nm}$ (use subscript for EX and EM; add a space between numbers and the units)

RESPONSE: We would like to thank the reviewer for their kind efforts to notice and inform us about these stylistic errors. As advised, we have gone through our manuscript with better scrutiny to locate and fix any such errors.

REVIEWERS' COMMENTS:

Reviewer #1 (Remarks to the Author):

The authors have completed a substantial body of new work demonstrating the utility and some important limitations of their new tool for quantifying ethylene in plant tissues. Despite remaining uncertainties, the present level of AEP sensor characterisation is satisfactory for an initial publication. The sensor limitations do somewhat diminish the importance of this work for plant molecular biologists, for example those that work on Arabidopsis. Nonetheless, it does seem that the AEP sensor could still be used in plant biology and agronomic studies of, for example, fruit ethylene production rates and spatial distributions.

Minor comment - the authors make the claim that the probe will not pass the cell wall and therefore will remain extracellular. Perhaps plasma membrane is meant?

Reviewer #2 (Remarks to the Author):

The revised manuscript is a significantly improved in my opinion. The authors have taken into consideration the reviewers comments and have done many additional experiments. They have done a great effort in setting up in vitro experiments with pure ethylene gas to test sensor selectivity, sensitivity and responsiveness (over time). The latter experiments (although not perfect due to the difficulties of working with ethylene gas) are critical to demonstrate sensor characteristics and potential. They have also done substantial new experiments in plants to show the spatiotemporal power of the AEP sensor. I like their new fruit data (slices of kiwi, pear, apple, grape, carrot and bell pepper) over the previous data (mashed apple tissue). Altogether, the new data presented makes the potential applicability and usability of this new AEP sensor more realistic. Despite some limitations of the AEP sensor (now clearly mentioned in the text), it is a major breakthrough in the field of plant biology. In my opinion, this AEP sensor is a unique tool that can change the field of ethylene biology. I look forward to the new discoveries made using this AEP sensor (or future improved versions) for ethylene detection in plants.

Please correct the following minor comments:

- L202: this should be tomato (and likely other climacteric fruit)
- L257: Replace "Through expression of the ethylene-overproduction protein 1 (ETO1), ACS activity can be positively regulated by preventing proteasome-dependent degradation. This is based on a crucial interaction between the C-terminal tail of type II ACS and ETO1. " by "The eto1-1 mutation results in the stabilization of type II ACS, preventing proteasomal degradation of ACS."

Consider the following comments:

- The dose-response curve developed using THF solvent showed that the RuQ sensor has only a limit of detection 27 ppm, which is levels that plants will never produce or encounter during physiologically relevant processes. Maybe mention this?
- The 167 ppm used for the AEP shows a time-dependent reactivity towards ethylene, reaching a linear fluorescence increase after roughly 60 min. This experiment is very valuable to show the reactivity, yet the concentrations used (167 ppm) are unrealistic for plant applications. A dilution curve in ethylene could have been tested (in their balloon set-up; which I like) using an inert gas (such as Hydrogen or helium) which is even less soluble comparable to ethylene gas. The partial pressure of each gas can then be used to calculate ethylene solubility using Henry's law. Preferentially, the lower ppm and even ppb levels should have been tested to make it relevant of studies in plants.

I appreciate the following:

- The spatial imaging experiment with kiwi fruit is beautiful. It really shows the potential of the sensor.

- I like the new limitations section (L345). But I would downsize the limitations regarding fluorescence microscopy. It is worth mentioning that good focal planes are difficult to obtain, but experienced cell biologist will be able to make plant sample preparations that are nice in one plane or use confocal microscopy. Also it is possible to use several fixing techniques to make nice slices of tomato fruit (which the authors could not do). So I feel this limitation is not a general one, but rather a limitation of the author's instrumentation/slice-making-capacity.

Reviewer #3 (Remarks to the Author):

The authors have taken great care in addressing the reviewers' comments and performed the requested additional experiments, in particular spatial imaging and time-dependent reactivity and imaging studies. Hence, the manuscript is much improved. The determination of the LOD of the probe proved challenging, but the limitations of the AEP probe are now clearly described. Considering the novelty of the approach, the manuscript should now be acceptable for publication in Nature Communications.

On behalf of my colleagues and myself, we would like to take this opportunity to thank all reviewers for their constructive criticism and kind feedback. Without a doubt, our manuscript has improved significantly due to your crucial guidance. The following are the responses to remaining comments. Thank you again for your invaluable time, input, and assistance.

----- *Reviewer 1* -----

1 COMMENT: The authors make the claim that the probe will not pass the cell wall and therefore will remain extracellular. Perhaps plasma membrane is meant?

RESPONSE: We thank the reviewer for the correction. As advised, we have changed the term 'cell wall' to 'cell membrane'.

----- *Reviewer 2* -----

2 COMMENT: L202: this should be tomato (and likely other climacteric fruit)

RESPONSE: In response to this request, 'fruits like apples' was changed to 'tomatoes (and likely other climacteric fruit)'.

3 COMMENT: L257: Replace "Through expression of the ethylene-overproduction protein 1 (ETO1), ACS activity can be positively regulated by preventing proteasome-dependent degradation. This is based on a crucial interaction between the C-terminal tail of type II ACS and ETO1." by "The eto1-1 mutation results in the stabilization of type II ACS, preventing proteasomal degradation of ACS."

RESPONSE: We thank the reviewer for the suggestion. As advised, we have changed the wording of that section to align with what was suggested.

4 COMMENT: The dose-response curve developed using THF solvent showed that the RuQ sensor has only a limit of detection 27 ppm, which is levels that plants will never produce or encounter during physiologically relevant processes. Maybe mention this?

RESPONSE: As suggested by the reviewer, following the mention of the RuQ sensor LOD, we have added a phrase comparing it to physiological ethylene concentrations.

5 COMMENT: The 167 ppm used for the AEP shows a time-dependent reactivity towards ethylene, reaching a linear fluorescence increase after roughly 60 min. This experiment is very valuable to show the reactivity, yet the concentrations used (167 ppm) are unrealistic for plant applications. A dilution curve in ethylene could have been tested (in their balloon set-up; which I like) using an inert gas (such as Hydrogen or helium) which is even less soluble comparable to ethylene gas. The partial pressure of each gas can then be used to calculate ethylene solubility using Henry's law. Preferentially, the lower ppm and even ppb levels should have been tested to make it relevant of studies in plants.

RESPONSE: We would like to thank the reviewer for bringing up this discussion. During our experimental planning, we also thought of proceeding with the experiment as described. However, we encountered practical limitations that greatly complicated the experiments.

In order to get a dose response curve with more than 2 points, several points need to be run where the hydrogen balloon is set at a larger volume than the ethylene balloon. For example, points can be done with the ethylene/hydrogen balloon ratio at 1:1, 1:2, 1:3, 1:4, and so on.

For 1:1, two balloons of equal size are easy enough to eyeball. However, for the other balloon sizes, it became difficult with our experimental setup to quantitatively ensure the balloon was filled with an accurate volume of gas (note: since balloons are not perfect spheres, measuring the diameter to find out the volume was also not viable).

As an alternative, we also thought about using multiple hydrogen balloons of the same size. For example, we could attach 4 hydrogen balloons (of equal size to the ethylene balloon) instead of

filling one hydrogen balloon with 4 times the volume. However, since we are using a cuvette capped by a septum, we noticed multiple needle punctures lead to gas leakage over the 1 hour incubation time.

Additionally, we encountered an instrumentation problem where our spectrofluorometer can only accommodate the cuvette attached to one balloon at a time. Any more balloons prevent the lid from properly closing.

Overall, we really do understand the request for us to obtain a proper dose response curve for our **AEP** probe. As it stands, however, we currently do not see a clear path to complete this without additional resources (new spectrofluorometer) and time. For that, we hope the reviewer can kindly understand our predicament.

6 COMMENT: I would downsize the limitations regarding fluorescence microscopy. It is worth mentioning that good focal planes are difficult to obtain, but experienced cell biologist will be able to make plant sample preparations that are nice in one plane or use confocal microscopy. Also it is possible to use several fixing techniques to make nice slices of tomato fruit (which the authors could not do). So I feel this limitation is not a general one, but rather a limitation of the author's instrumentation/slice-making-capacity.

RESPONSE: After consideration of the reviewer's comments, we agree with their assessment that our issues with preparing samples for fluorescence microscopy is not a general limitation. As such, we have removed the part of the text describing this problem so not to conflict with readers who are more experienced cell biologists.

----- *Reviewer 3* -----

N/A